# WEB AGENTS WITH WORLD MODELS: LEARNING AND LEVERAGING ENVIRONMENT DYNAMICS IN WEB NAVIGATION

**Hyungjoo Chae**   **Namyoung Kim**   **Kai Tzu-iunn Ong**   **Minju Gwak**
**Gwanwoo Song**   **Jihoon Kim**   **Sunghwan Kim**   **Dongha Lee**   **Jinyoung Yeo**

Yonsei University

{mapoout, namyoung.kim, donalee, jinyeo}@yonsei.ac.kr

 https://github.com/kyle8581/WMA-Agents
 https://hf.co/spaces/LangAGI-Lab/WMA-Agents

## ABSTRACT

Large language models (LLMs) have recently gained much attention in building autonomous agents. However, performance of current LLM-based web agents in long-horizon tasks is far from optimal, often yielding errors such as repeatedly buying a non-refundable flight ticket. By contrast, humans can avoid such an irreversible mistake, as we have an *awareness* of the potential outcomes (*e.g.*, losing money) of our actions, also known as the "*world model*". Motivated by this, our study first starts with preliminary analyses, confirming the absence of world models in current LLMs (*e.g.*, GPT-4o, Claude-3.5-Sonnet, etc.). Then, we present a World-model-augmented (WMA) web agent, which simulates the outcomes of its actions for better decision-making. To overcome the challenges in training LLMs as world models predicting next observations, such as repeated elements across observations and long HTML inputs, we propose a transition-focused observation abstraction, where the prediction objectives are free-form natural language descriptions exclusively highlighting important state differences between time steps. Experiments on WebArena and Mind2Web show that our world models improve agents' policy selection without training and demonstrate superior cost- and time-efficiency compared to recent tree-search-based agents.

## 1 INTRODUCTION

Large language models (LLMs) have been widely applied to solve tasks in diverse domains, including web navigation, where LLMs generate action sequences (*e.g.*, click) to accomplish user goals on websites (Shi et al., 2017; Kim et al., 2024). Despite some success (Yao et al., 2022), LLM-based web agents' performance remains significantly poor in long-horizon environments such as WebArena (Zhou et al., 2023), where GPT-4 yields a task success rate of 14.41% whereas humans have a success rate of 78.24%. This raises a question: *Why do LLMs, despite their advancements, perform much worse than humans in web navigation?*

Humans avoid unwanted situations by considering the possible outcomes of our actions beforehand (Edwards, 1954). Such awareness of actions and outcomes is referred to as the "*world model*" (Forrester, 1995). Meanwhile, existing LLM-based web agents rely heavily on trial and error to make decisions, as they lack world models to help them foresee the outcome of an action without actually performing it (LeCun, 2022), leading to sub-optimal decision-making that is irreversible (*e.g.*, repeatedly buying a non-refundable item). Acknowledging the importance of world models, studies in robotics and reinforcement learning (RL) have proposed to incorporate world models for agents in navigation tasks. For instance, Du et al. (2023) and Yang et al. (2024) apply

world models to simulate visual outcomes/observations of input texts or robot control. The Dreamer series use world models to predict latent state of images and use them to optimize policies, reducing the need for actual interactions in game environments (Hafner et al., 2019a; 2020; 2024).

Motivated by these, this paper begins by investigating SOTA LLMs' understanding of "environment dynamics", *i.e.*, the association between actions and environment states. We reveal that (i) current LLMs (*e.g.*, GPT-4o and Claude-3.5-Sonnet) struggle with predicting the outcomes of their actions and (ii) the awareness of potential outcomes helps them make decisions aligning with user goals. Upon these findings, we present a World-Model-Augmented (WMA) web agent, which simulates the outcomes of its actions for better decision-making. However, naively training a world model to predict the next observation state (*i.e.*, the entire webpage) can lead to a large amount of repeated elements across observations and long HTML inputs, negatively affecting model performance. Thus, we propose a novel transition-focused observation abstraction, where the world model is trained to generate free-form natural language descriptions exclusively highlighting important state differences between time steps (*e.g.*, an updated price on the website). During inference, our agent first simulates the outcome (*i.e.*, next observation) of each action candidate (from the policy model) using the world model. Then, a value function estimates the rewards of all simulated observations, helping the agent select a final action with the highest estimated reward. Our contributions are two-fold:

- We are the first to pioneer world models in LLM-based web agents, laying the groundwork for policy adaptation through simulated environment feedback in web navigation.
- We present a novel transition-focused observation abstraction for training LLMs as world models. We show that using world models trained with this method can improve action selection by simulating the action candidates without training the policy models. Also, we demonstrate our agents' cost- and time-efficiency compared to recent tree-search-based agents (Koh et al., 2024), by 6.8x and 5.3x, respectively.

## 2 RELATED WORK

**Benchmarks for web agents.**  Many benchmarks have been introduced to evaluate LLM-based agents' ability in web navigation (Kim et al., 2024). MiniWoB (Shi et al., 2017) and Mini-WoB++ (Liu et al., 2018) are among the first widely adopted benchmarks. More recently, Web-Shop (Yao et al., 2022) simulates e-commerce environments where agents are tested to execute tasks based on text instructions on the web. These early benchmarks are limited to specific and constrained domains. Mind2Web (Deng et al., 2024) curates web tasks across more diverse domains, and WebArena (Zhou et al., 2023) emphasizes functional correctness and more realistic scenarios (*e.g.*, posting articles on Reddit) in simulated environment. We adopt Mind2Web and WebArena for evaluation for their generalizability and resemblance of real-world web interactions.

**LLM-based web agents.**  In recent years, LLM-based agents have become popular in the web navigation domain. However, since many powerful proprietary LLMs do not provide access to model parameters, many studies of web navigation have been focusing on training-free methods where LLMs directly learn from user inputs (*i.e.*, prompts) without task-specific training (Sodhi et al., 2023; Zheng et al., 2023). For instance, Wilbur (Lutz et al., 2024) and Agent Workflow Memory (Wang et al., 2024b) leverage a verification model (Pan et al., 2024) with prompt-based methods to collect successful trajectory data for guiding the agent's policy at inference time. AutoEval (Pan et al., 2024) and Tree search agent (Koh et al., 2024) increase the number of trials and reasoning paths, further improving system performance. However, due to their trial-and-error nature, these approaches can not only be computationally inefficient in gathering trajectories as tasks become more complex but also are more prone to undesired results (*e.g.*, booking a non-refundable ticket). Our WMA web agent reduces such risks via a *world model*, which predicts future observations and the rewards of their corresponding action candidates before actually making an action. Furthermore, our approach can be orthogonally applied to many of the existing methods.

**World models in autonomous agents.**  *World models* refer to systems that generate internal representations of the world, predicting the effects of their actions on environments (LeCun, 2022). In RL, simulating observations and environmental feedback using world models allow the policy model to learn (Sutton, 1990) or plan (Ha & Schmidhuber, 2018; Hafner et al., 2019b) without

actually interacting with the environment. While some world models are trained with raw observations (Oh et al., 2015; Chiappa et al., 2017), others are built on latent representations (Hafner et al., 2019a; 2020; Kipf et al., 2020). For instance, in the image domain, Hafner et al. (2020) train a world model by training it to first compute a posterior stochastic state based on the current image and then a prior stochastic state that tries to predict the posterior without access to the image. Within the field of LLMs, Zhang et al. (2024) convert visual observations into natural language and employs an LLM-based world model for text-based games, and Wang et al. (2024a) further transform observations into a structural format (*e.g.*, JSON), improving LLMs' reasoning over state transition functions. In web navigation, environments are built upon not only natural language but on more complex text modalities such as HTML and DOM trees. We address this by transforming them to a novel free-form description, highlighting the state difference between each time step.

## 3 PRELIMINARY ANALYSES: ARE CURRENT LLMS AWARE OF ENVIRONMENT DYNAMICS IN WEB NAVIGATION?

We first start with investigating whether LLMs can understand the association between actions and their effects on the environment, *i.e.*, understand the *environment dynamics*. We conduct analyses addressing these two questions:

- **Preliminary question I**: *Are LLMs aware of the outcomes of their actions?*
- **Preliminary question II**: *When having access to the outcome of each action candidate, can LLMs select an optimal action aligning with the user objective?*

For the analyses, we sample 100 user instructions from WebArena and annotate human trajectories within the environment. Each instance has a user instruction, the current state, a human-annotated golden action, and the corresponding next state resulting from the golden action. We analyze 4 popular closed-source SOTA LLMs: GPT-4o-mini (Zhu et al., 2023), GPT-4o, GPT-4-Turbo (OpenAI, 2023), and Claude-3.5-Sonnet (Anthropic, 2024). More details are in Appendix B.

### 3.1 PRELIMINARY ANALYSIS I - LLMS STRUGGLE WITH PREDICTING THE NEXT STATES CAUSED BY THEIR ACTIONS

**Setups.** We test LLMs' ability to predict the outcomes of actions on the web via a binary classification task. Given the current state and the golden action, the LLM is prompted to select the correct next state from (i) the golden next state and (ii) a lexically similar yet incorrect next state retrieved from the same trajectory. We calculate the lexical similarity with difflib (Python, 2024). We assess classification accuracy.

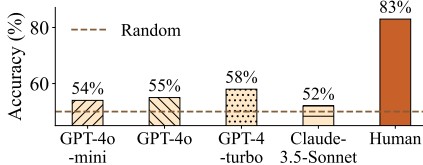

Figure 1: LLMs' performance in next state prediction.

**Results.** Figure 1 reveals that under vanilla settings, current LLMs cannot effectively predict the next states caused by their actions. First, all adopted LLMs (54.75% on average) lose significantly to humans. Also, Claude-3.5-Sonnet performs almost as badly as random guessing. These suggest that the world model, the ability to foresee the potential outcomes of actions taken, is absent in LLMs.

### 3.2 PRELIMINARY ANALYSIS II - LLMS MAKE BETTER ACTION SELECTION WHEN ACCESSING THE OUTCOME OF EACH ACTION CANDIDATE

**Setups.** We assess whether LLMs can select a correct action that aligns with the user goal when they are provided with the outcome of each action candidate. Given the current state, 10 action candidates, and their corresponding outcomes/next states, the LLM is prompted to differentiate the golden action from other 9 negative actions.

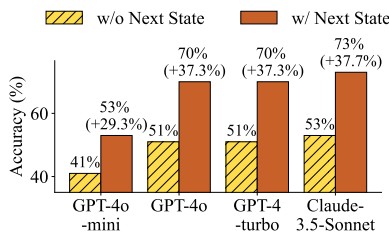

Figure 2: LLMs' performance in action selection (w/ and w/o next states).

**Results.** Figure 2 compares LLMs' performance in differentiating the golden action from negative actions when

**World Model Training**

**Step I: Harvesting Agent Trajectories**

Policy model $\xrightarrow{\text{Action } a}$ $\xleftarrow{\text{Observation } O}$ Environment

$\{I, o_t, a_t, o_{t+1}\}$

Agent-environment interaction data

**Step II: Transition-focused Observation Abstraction**

State transition $\qquad$ Abstracted observation

$\Delta(o_t, o_{t+1}) \longrightarrow$ (LLM) $\longrightarrow \tilde{o}_{t+1}$

UPDATED: [120] StaticText
DELETED: [131] Button...
ADDED: ...

The effects of the action are ...

**Step III: Learning Environment Dynamics**

$\{I, o_t, a_t, o_{t+1}\}$
$\qquad$ *updating*
$\{I, o_t, a_t, \tilde{o}_{t+1}\}$

World Model $\phi$

Training data $\tilde{\mathcal{D}}$

**Inference-time Policy Optimization via the World Model**

$o_t$
$\downarrow$
Policy model ❄

Action candidates
$\begin{Bmatrix} a_t^1 \\ a_t^2 \\ \vdots \\ a_t^k \end{Bmatrix}$
$\rightarrow$
World Model $\phi$
*Predicting the next observation*
$\rightarrow$
Predicted next observations
$\begin{Bmatrix} \tilde{o}_{t+1}^1 \\ \tilde{o}_{t+1}^2 \\ \vdots \\ \tilde{o}_{t+1}^k \end{Bmatrix}$
$\rightarrow$
Value Function $V$
$\rightarrow$
Estimated reward scores
$\begin{Bmatrix} 0.2 \\ 0.3 \\ \vdots \\ 0.1 \end{Bmatrix}$
$\xrightarrow{\text{argmax}}$
Selected action
$a_t^2$
*Selecting the action yielding the most optimal next state*

Figure 3: Framework overview. We first collect training data for world models (top). After training, we perform policy optimization by selecting the action leading to an optimal next state (bottom).

they are/are not provided with the resulting next state of each candidate action. We find that current SOTA LLMs have difficulty in selecting correct actions when they can only rely on the current observations/states (striped bars), yielding an average accuracy of only 49%. However, when augmented with the corresponding next state of each action candidate, they demonstrate huge performance gains (up to 38% improvement) in selecting correct actions. When only the current state and the user objective are provided, GPT-4o yields an accuracy of 53%. In contrast, when the next state is given, performance rises to 73%.

### 3.3 INSIGHTS FROM PRELIMINARY ANALYSES

Through our preliminary analyses, we have demonstrated that: **(i)** Web agents built with SOTA LLMs are bad at predicting how their actions affect next states; **(ii)** When being aware of how an action affects the next state, LLMs can make better decisions. These findings highlight the necessity of *world models* in LLM-based web agents, pointing out a promising direction for facilitating better web agents in complex, long-horizon navigation tasks.

## 4 WORLD-MODEL-AUGMENTED WEB AGENTS

Motivated by the above insights, we present a novel framework for World-Model-Augmented (WMA) web agents, LLM-based web agents equipped with *world models*. The world models learn/leverage environment dynamics (*i.e.*, association of actions and outcomes) to simulate plausible next observations of agents' actions, facilitating better decisions (*i.e.*, polices) in web navigation.

**Formulations.** Since web agents access only information in the viewport (*i.e.*, users' visible area), we model web navigation as a partially observable Markov decision process (POMDP). We consider a web environment $\mathcal{E}$ with: (i) a hidden state space $\mathcal{S}$; (ii) an action space $\mathcal{A}$, including language-guided actions (*e.g.*, CLICK, TYPE, HOVER, etc.) and their descriptions; (iii) an observation space $\mathcal{O}$ representing an accessibility tree of the page, which is a simplified DOM tree (Zhou et al., 2023).

In the POMDP, the agent receives a new partial observation $o_{t+1} \in \mathcal{O}$ from $\mathcal{E}$ after performing an action $a_t \in \mathcal{A}$ based on $o_t$. Such state transition $s_t \rightarrow s_{t+1}$ is managed by a golden transition function $\mathcal{T} : \mathcal{S} \times \mathcal{A} \rightarrow \mathcal{S}$ provided in the environment.

### 4.1 WORLD MODEL TRAINING

We hereby introduce the training process of our world models. As shown in Figure 3 (top), our training consists of three main steps:

### 4.1.1 STEP I: HARVESTING AGENT-ENVIRONMENT INTERACTION DATA

We start by collecting the dataset $\mathcal{D} = \sum_{t=1}^{n}\{I, o_t, a_t, o_{t+1}\}$ from the environment $\mathcal{E}$ for training world models. For that, we prompt an LLM as web agent to achieve the goal provided in the user instruction $I$, by iteratively predicting an action $a_t$ based on the current observation $o_t$ throughout all $n$ time steps. Consequently, we obtain $\mathcal{D}$ from trajectory $\tau = \{o_1, a_1, o_2, ..., a_n, o_{n+1}\}$ based on $I$, and environment states of $n$ time steps $\{s_1, ..., s_{n+1}\} \subset \mathcal{S}$ obtained via transition function $\mathcal{T}$.

### 4.1.2 STEP II: TRANSITION-FOCUSED OBSERVATION ABSTRACTION

With the collected data $\mathcal{D} = \sum_{t=1}^{n}\{I, o_t, a_t, o_{t+1}\}$, it is intuitive to train LLM-based world models to predict $o_{t+1}$, which is expressed with texts (*e.g.*, HTML and accessibility tree) (Deng et al., 2024; Zhou et al., 2023). However, simply using textual observations to represent environment states and use them as training objectives may introduce the following downsides:

- **Low information gain during training**: State transitions in websitesoften involve altering only a part of the previous observation (*e.g.*, a drop-down menu is clicked). As a result, most information in $o_{t+1}$ remains the same as it is in $o_t$. Therefore, predicting the entire textual observation from scratch may result in low information gain during training.

- **Excessively long sequence length**: Processing the whole text-based observations can lead to excessively long sequence length and consequently high computational costs. Indeed, this can be partially mitigated by replacing raw HTML with an accessibility tree (relatively simple), using it as LLMs' training objectives still introduce a long sequence length (4K tokens on average, see Figure 4).

To address the above bottleneck in training text-based models (*i.e.*, LLMs) as world models, we draw inspiration from how the RL community conventionally implements world models: using estimated latent vectors as summaries of raw visual observations, reducing memory footprints for effectively learning environment dynamics (Doerr et al., 2018; Hafner et al., 2019a) – We thus propose to abstract raw text observations, with a focus on state transition between consecutive observations, for obtaining better training objectives.

Figure 4: Sequence length distribution of different observation representations.

To collect abstracted next observations for training world models, one may simply run an off-the-shelf summarizer on $o_{t+1}$ collected in Step I. However, while reducing sequence length, this does not address the low information gain caused by repeated elements between $o_t$ and $o_{t+1}$. Thus, instead of such a naive approach, as shown in Figure 5, we first (i) apply the Hungarian algorithm (Kuhn, 1995) to calculate a cost matrix for matching elements between $o_t$ and $o_{t+1}$ and (ii) mechanically transform the results into a list of state transition $\Delta(o_t, o_{t+1})$, pointing out UPDATED, DELETED, and ADDED elements on the web. After that, we prompt an LLM to convert the extracted $\Delta(o_t, o_{t+1})$ into a free-from natural language description $\tilde{o}_{t+1}$, which highlights the difference between the new observation $o_{t+1}$ and $o_t$. Replacing $o_{t+1}$ in $\mathcal{D} = \{I, o_t, a_t, o_{t+1}\}$ collected in Step I with $\tilde{o}_{t+1}$ we just acquired here, we get a final dataset $\tilde{\mathcal{D}} = \sum_{t=1}^{n}\{I, o_t, a_t, \tilde{o}_{t+1}\}$ for training world models.

### 4.1.3 STEP III: LEARNING ENVIRONMENT DYNAMICS

Lastly, using $\tilde{\mathcal{D}}$, we proceed to train the internal world model $\phi$ of the web agent to learn the environment dynamics. Formally, an LLM working as the world model is trained to predict the abstracted observation $\tilde{o}$ of the next state $s_{t+1}$, given three inputs: the user instruction $I$, the current observation $o_t$, and the current action $a_t$. This LLM is trained to minimize the following loss term via the next-token prediction objective:

$$\mathcal{L}_\phi = -\log \sum_{(\tilde{o}, o, a, I) \in \tilde{\mathcal{D}}} p(\tilde{o}_{t+1} | o_t, a_t, I) \tag{1}$$

Through this process, this LLM learns the environment dynamics, working as a world model that helps the web agent to foresee the potential outcome (*i.e.*, predict the next observation) of its action.

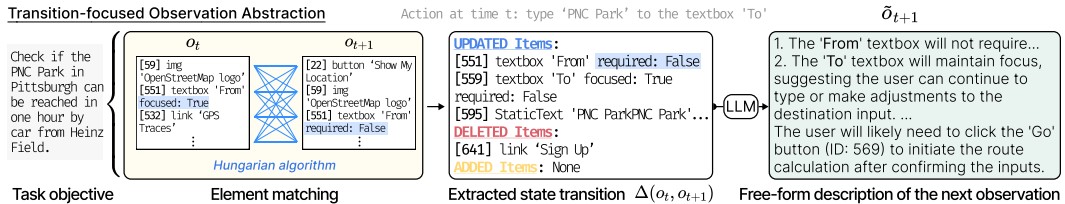

Figure 5: The overview of transition-focused observation abstraction.

## 4.2 INFERENCE-TIME POLICY OPTIMIZATION WITH THE WORLD MODEL

In this section, we explain how we use the developed world model $\phi$ to improve LLM-based agents' performance in web navigation. As illustrated in Figure 3 (bottom), the web agent consists of three main components: (i) a policy model $\theta$; (ii) our world model $\phi$; (iii) a value function $V$. Note that the policy model $\theta$ is frozen, *i.e.*, we do not update its parameters.

During inference at time $t$ with a current observation $o_t$, WMA web agent ought to utilize the world model $\phi$ to foresee how an action can affect the state (*i.e.*, predict $\tilde{o}_{t+1}^i$), and accordingly find an optimal action/policy $a_t$ from the policy model $\theta$ that can lead to the target goal defined in $\mathcal{I}$.

We begin by sampling $k$ action candidates $\{a_t^1, a_t^2, ..., a_t^k\}$ from $\theta$ via top-$p$ decoding (Holtzman et al., 2019), to conduct diverse exploration on future observations $\{o_{t+1}^1, o_{t+1}^2, ..., o_{t+1}^k\}$ similar to Koh et al. (2024). Next, we use the world model $\phi$ to "*simulate*" the potential next observation $\tilde{o}_{t+1}^i$ caused by each action candidate $a_t$:

$$\{\tilde{o}_{t+1}^i\}_{i=1}^k = \{\phi(o_t, a_t^i, I)\}_{i=1}^k \tag{2}$$

Note that each $\tilde{o}_{t+1}^i$ is a free-form description of the next observation, as shown in Figure 5 (right).

Lastly, we decide the agent's action for actual operation by selecting the action leading to the most optimal future state $s_{t+1}$ from all action candidates. Following Koh et al. (2024), we use an off-the-shelf LLM as a value function $V(\cdot)$ to estimate the reward yielded by each action candidate and select the action $\hat{a}_t$ with the highest reward:

$$\hat{a}_t = \underset{a_t \in \{a_t^1, ..., a_t^k\}}{\operatorname{argmax}} V(I, o_t, a_t, \tilde{o}_{t+1}^i) \tag{3}$$

With this process, we are able to optimize the policy selection of web agents in inference time without training the policy models. This training-free augmentation of world models allows us to easily adapt our world model $\phi$ to existing web agents, including prompt-based (Pan et al., 2024; Wang et al., 2024b) and fine-tuned LLMs (Gur et al., 2023; Lai et al., 2024).

## 5 EXPERIMENTS

### 5.1 SETUPS AND IMPLEMENTATION DETAILS

**Benchmarks and evaluation metrics.** For evaluation, we use the official WebArena and Mind2Web benchmarks (Zhou et al., 2023; Deng et al., 2024). WebArena includes 812 real-life tasks in simulated environments across five different websites, spanning four key domains - e-commerce (Shopping), social forums (Reddit), collaborative software development (Gitlab), content management (CMS), and Map. Details of each domain are further explained in Appendix C.3. The main metric, Success Rate (SR), is calculated as the percentage of the user instructions that are successfully accomplished by the generated agent trajectory. On the other hand, Mind2Web (Deng et al., 2024) covers over 2,000 open-ended tasks, collected from 137 websites of 31 domains and crowd-sourced action sequences for the tasks. Along with the SR, Mind2Web also uses Step SR, which measures whether the predicted action selects both the correct action type (action $F_1$) and element ID (element accuracy). When the agent succeeds in all steps in a trajectory, it is evaluated as success.

Table 1: Agent performance in WebArena. Δ: relative performance gains from policy optimization.

| Policy LLMs | Methods | Max Actions | Success Rate (SR) | | Δ |
| --- | --- | --- | --- | --- | --- |
| | | | w/o Action Selection | w/ Action Selection | |
| GPT-4 | AutoEval (Pan et al., 2024) | 30 | 20.2% | - | - |
| | BrowserGym (Drouin et al., 2024) | | 23.5% | - | - |
| | SteP (Sodhi et al., 2023) | | 35.8% | - | - |
| | AWM (Wang et al., 2024b) | | 35.5% | - | - |
| GPT-4o | Vanilla CoT (Zhou et al., 2023) | 30 | 13.1% | - | - |
| | Tree search agent (Koh et al., 2024) | 5 | 15.0% | 19.2% | +28.0% |
| | **WMA web agent (ours)** | 5 | 12.8% | 16.6% | +29.7% |
| GPT-4o-mini | **WMA web agent (ours)** | 5 | 9.4% | 13.5% | +43.6% |

Table 2: Domain-specific performance of agents using GPT-4o-mini as policy models

| Methods / Domains | Shopping | CMS | Reddit | Gitlab | Map | Overall |
| --- | --- | --- | --- | --- | --- | --- |
| Vanilla CoT (max actions = 5) | 18.8% | 8.2% | 5.3% | 3.1% | 11.6% | 9.4% |
| WMA web agent (ours) | 19.3% | 11.5% | 7.9% | 8.7% | 22.3% | 13.5% |
| Δ | +3% | +40% | +49% | +181% | +92% | +44% |

**Training data for world models.** (i) For evaluation in WebArena: To facilitate applications in the real world, the training data for world models needs to cover a wide range of tasks/goals. Since a diverse and large-scale user instructions set is not available,[1] we synthesize user instructions using an LLM. With these synthesized instructions of various goals, we are able to collect rich trajectories as training data, improving world models' generalization to diverse real-world situations. In practice, we sample $l$ trajectories for each $I$.[2] We generate 870 synthetic user instructions and gather 14K instances from WebArena using GPT-4o-mini as the policy model. To avoid redundant learning, we filter out repeated state-action pairs. (ii) For evaluation in Mind2Web: we adopt the offline trajectory data from Mind2Web, following the setting of Wang et al. (2024b).

**Baselines.** For baselines, we adopt: (1) a prompting-based LLM (Zhou et al., 2023) powered by chain-of-thought prompting (Wei et al., 2022); (2) AutoEval (Pan et al., 2024). It refines agents' trajectories based on the feedback on the final state of the trajectory (*i.e.*, *succeed* or *fail*) from a VLM evaluator (Shinn et al., 2024); (3) BrowserGym (Drouin et al., 2024) trains web agents with multi-modal observations, including HTML contents and the screenshot image of the browser; (4) SteP (Sodhi et al., 2023), a framework based on human-authored hierarchical policies injected to the agent; (5) HTML-T5 (Gur et al., 2023), the previous SOTA method on Mind2Web, uses LLMs pre-trained LLMs on HTML corpus. (6) Agent workflow memory (AWM;Wang et al. (2024b)) leverages self-discovered workflow memory to guides its policy; (7) Tree search agent (Koh et al., 2024), the most competitive baseline that explores multiple trajectories and selects an optimal path via a tree search algorithm. The main difference between ours and Tree search agents is that ours only uses the predicted future states via simulation and does not actually explore diverse states.

**World model.** We use Llama-3.1-8B-Instruct (Dubey et al., 2024) as the backbone LLM for our world models.[3] For WebArena, we construct our dataset in online setting using the provided web environment. In Mind2Web, we use the offline trajectory data (*i.e.*, the train set) following Wang et al. (2024b). For prompt-based world models (baselines) in our experiments, we use 2-shot demonstrations to instruct LLMs to predict the next state. More details are provided in Appendix C.1.

**Policy model.** Following Koh et al. (2024), we adopt GPT-4o (gpt-4o-0513) as the agent backbone for evaluation in WebArena. Additionally, we test with GPT-4o-mini (gpt-4o-mini-0718) to test our framework in relatively more resource-restricted scenarios.

---

[1]In WebArena, only test data (*i.e.*, instructions) is provided.

[2]We empirically set $l = 5$ in our work. Further details on the whole data collection process are in Appendix.

[3]https://huggingface.co/meta-llama/Meta-Llama-3.1-8B-Instruct

Table 3: Success rate on Mind2Web tests using GPT-3.5-Turbo as policy models. EA = element accuracy; EF = element filtering; $AF_1$ = action $F_1$; * = results from the original paper.

| Methods | Cross-Task | | | | Cross-Website | | | | Cross-Domain | | | |
|---|---|---|---|---|---|---|---|---|---|---|---|---|
| | EA | $AF_1$ | Step SR | SR | EA | $AF_1$ | Step SR | SR | EA | $AF_1$ | Step SR | SR |
| Synapse* | 34.4% | - | 30.6% | 2.0% | 28.8% | - | 23.4% | 1.1% | 29.4% | - | 25.9% | 1.6% |
| HTML-T5-XL* | 60.6% | **81.7%** | 57.8% | 10.3% | 47.6% | 71.9% | 42.9% | 5.6% | 50.2% | **74.9%** | 48.3% | 5.1% |
| MindAct* | 41.6% | 60.6% | 36.2% | 2.0% | 35.8% | 51.1% | 30.1% | 2.0% | 21.6% | 52.8% | 18.6% | 1.0% |
| AWM (w/ EF)* | 50.6% | 57.3% | 45.1% | 4.8% | 41.4% | 46.2% | 33.7% | 2.3% | 36.4% | 41.6% | 32.6% | 0.7% |
| AWM (w/o EF) | 78.3% | 74.1% | 62.8% | 15.3% | 74.7% | 70.1% | 58.6% | 6.2% | 74.8% | 71.2% | 60.7% | 9.5% |
| AWM+WMA (ours) | **79.9%** | 75.8% | **67.0%** | **25.4%** | **75.7%** | **72.1%** | **61.3%** | **8.5%** | **75.9%** | 72.6% | **63.4%** | **10.1%** |

**Value function.** We fine-tune Llama-3.1-8B-Instruct to predict rewards using data from Mind2Web, where rewards (as training objective) are calculated based on the progress toward the goal, *i.e.*, $t/(len(\tau))$ when $a_t$ is taken.

## 5.2 MAIN RESULTS

**Agent performance in WebArena.** In Table 1 (middle), we first compare our WMA web agent (16.6%) with vanilla CoT (13.1%) and observe significant improvements over almost all domains in WebArena as detailed in Table 2. Interestingly, when using GPT-4o-mini as the policy model, our agent achieve 181% and 92% performance gains over CoT in Gitlab and Map, respectively. The relatively small improvement in Shopping might be due to the large-scale state space $\mathcal{S}$ in the domain, such as the diversity of searched item lists from different user queries, which makes it harder for the world model to properly learn environment dynamics. Regardless, the overall improvement suggests the effectiveness of leveraging learnt environment dynamics during inference time.

Next, we compare our approach with Tree search agent (Koh et al., 2024), which uses the oracle next state observation (*i.e.*, resulted by the gold transition function $\mathcal{T}$ from the environment) for policy selection instead of estimated observation via the world model. While the absolute SR of our WMA agent (16.6%) is slightly below Tree search agent (19.2%) when using GPT-4o as policy models, our policy optimization with the world model brings a larger performance gain to vanilla CoT than tree search (+29.7% vs. +28.0%). Also, in the later section (§5.3), we present ours' superior cost and time efficiency over Tree search agent.

**Agent performance in Mind2Web.** We compare WMA web agent with MindAct (Deng et al., 2024) and AWM (Wang et al., 2024b), which are previous and current SOTAs on Mind2Web.[4] Table 3, demonstrates that WMA web agent significantly outperforms AWM,[5] achieving new SOTA performance. Furthermore, the results indicate that WMA web agent trained on Mind2Web data has a strong generalization capability. This makes our approach much more valuable in scenarios where collecting data for new web environments is non-trivial.

**Our advantages besides performance gains.** Based on the performance reported, we can conclude that our strategy of building world models (*i.e.*, observation abstraction) is effective not only for accessibility tree format (WebArena) but also for HTML format (Mind2Web), underscoring the applicability of our approach across different representations of web data. Another advantage of our approach over others is that the developed world models can be incorporated into existing or future web agents without any additional training of policy models, enabling easy implementation.

## 5.3 ANALYSES OF TIME AND COST EFFICIENCY

We compare our WMA web agent with Tree search agent in terms of time and API cost efficiency. Results are shown in Table 4. To run one user instruction, Tree search agent spends about 748.3 seconds on average, as it involves the exploration of diverse future states while actually interacting with the environment. When it conducts backtracing to revert to the previous state, the whole sequence of previous actions has to be executed again. By contrast, WMA web agent only takes 140.3 seconds per instance by simulating the possible action candidates rather than actually executing them, which

---

[4] Tree search agent is not applicable to this benchmark as the environment is not available.

[5] Surprisingly, we find element filtering (EF) of MindAct, applied to AWM in default, largely hindering its performance. Thus, in Table 3, we include the results without EF. A detailed discussion is in Appendix C.6

Table 4: Head-to-head comparison of Tree search agent (results are from Koh et al. (2024)) and ours regarding (i) SR and (ii) API cost, and (iii) inference time. We use GPT-4o for policy models.

| Methods | Shopping | CMS | Reddit | Gitlab | Map | API cost | Inference time (sec) |
|---------|----------|-----|--------|--------|-----|----------|----------------------|
| Tree search agent | 28.1% | 16.5% | 10.5% | 13.3% | 25.8% | $2.7 | 748.3 |
| WMA (ours) | 20.8% | 14.3% | 10.5% | 13.3% | 26.8% | $0.4 | 140.3 |

Table 5: Results of the ablation study in WebArena.

| Settings | World Model | | Success Rate (SR) | | | |
|----------|-------------|----------|-------------------|--------|-----|---------|
| | Use | Training | Shopping | Gitlab | Map | Overall |
| w/o next states in reward estimation (§4.2) | ✗ | ✗ | 28.0% | 6.0% | 19.0% | 18.0% |
| w/o training world models (§4.1) | ✓ | ✗ | 30.0% | 10.0% | 15.0% | 17.5% |
| w/o abstracting observations (§4.1.2) | ✓ | ✓ | 22.0% | 6.0% | 15.0% | 14.5% |
| WMA (ours) | ✓ | ✓ | 32.0% | 14.0% | 21.0% | 22.0% |

is 5.3 times faster than Tree search agent. Tree search agent requires 6.8 times more API cost due to its multi-modal inputs. To sum up, while showing comparable performance to Tree search agent in CMS, Reddit, Gitlab, and Map, our WMA web agent demonstrates superior cost and time efficiency.

## 5.4 ABLATION STUDIES

We conduct several ablation studies on our WMA web agent with 200 randomly sampled instances from WebArena (Shopping: 50; Gitlab: 50; Map: 100). We use GPT-4o-mini as policy models.

**Accessing simulated next states in reward estimation improves agent performance.** To assess the impact of incorporating the simulated next state when calculating the value score, we compare our reward estimation strategy to a Q-value function (Haarnoja et al., 2017) that predicts the reward based on only $(o_t, a_t)$. The results in Table 5 show that the information of the resulting next state helps the value function to predict rewards more accurately, resulting a better task performance.

**Fine-tuning facilitates better world models than prompt-based approaches.** To assess the effectiveness of our training approach for world models, we compare our framework with a variant, where we replace the trained world model (*i.e.*, fine-tuned Llama-3.1-8B-Instruct) with a GPT-4o-mini prompted to predict the next observation solely based on 2-shot demonstrations (*i.e.*, in-context learning) without training. The sub-optimal

Table 6: Performance with different value models.

| Value Function | Training | SR |
|----------------|----------|------|
| GPT-4o-mini | ✗ | 12.7% |
| Llama-3.1-8B | ✓ | 13.5% |

performance of this variant, as shown in Table 5 (2nd row), suggests that SOTA LLMs do not have sufficient knowledge of environment dynamics, which is consistent with our findings in §3.1.

**Abstracting observation elicits better next state prediction.** We evaluate the effectiveness of our observation abstraction (§4.1.2), which focuses on state transition. For that, we train a world model that learns to predict the full accessibility tree, *i.e.*, $o_{t+1}$ instead of our transition-focused abstraction $\tilde{o}_{t+1}$. As we expected, Table 5 (3rd row) reveals that generating the whole next observations (*i.e.*, all elements in the viewport) results indeed hinder agent performance, yielding the worst SR among all ablations. This shows that processing redundant and repeated in-

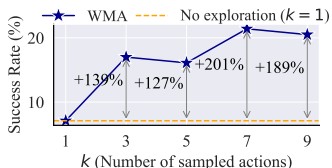

Figure 6: Ablation on the number of sampled actions ($k$).

formation across observations negatively affects the world model in capturing critical state changes compared to abstracted observations that exclusively highlight state transition.

**Choice of value functions.** We compare the fine-tuned value model (*i.e.*, Llama-3.1-8B-Instruct) used for implementing WMA web agents with prompted GPT-4o-mini in Table 6. Ours lead to a slightly better agent performance compared to GPT-4o-mini. This suggests fine-tuning the value function is a reasonable alternative in scenarios where API budgets are limited.

**Budget for exploration.** Figure 6 shows that there is a positive trend between the number of sampled actions ($k$) during inference-time policy optimization in §4.2) and the agents' task performance (SR). These results suggest that our WMA web agent may benefit from more exploration of the future states when the budget is allowed.

## 6 FURTHER ANALYSES

### 6.1 COMBINING SELF-REFINE WITH OUR WORLD MODELS

Besides our inference-time policy optimization, another way of using world models is to refine its predicted action (Madaan et al., 2024), based on the outcome simulated by the world model. Such self-refinement has been showing promising performance in diverse LLM applications (Shinn et al., 2024; Chae et al., 2024). Here, we conduct a demonstrative experiment of combining self-refine with our world model in the Map domain from WebArena. Since tasks in this domain involve a complex set

Table 7: Results of applying self-refine to GPT-4o-mini using simulated environment feedback.

| Methods | SR |
|---|---|
| Vanilla CoT | 11.6% |
| Self-refine w/ our world model | 13.4% |
| WMA (ours; Fig 3) | 22.3% |

of utility tools, such as sorting and zoom-in, we consider it suitable for testing self-refine. In this experiment, after the policy model $\theta$ produces an action $a_t$, we use our world model to simulate the next observation $\tilde{o}_{t+1}$ and prompt $\theta$ to refine the action based on $\tilde{o}_t$. Simply put, this setting allows $\theta$ to make adjustments to its output action when the predicted next observation is not optimal. Table 7 shows that refining with simulated environment feedback improves the agent's policy by 1.8% point in terms of accuracy compared to CoT. While this is a plausible direction for future work, our simulate-score-select paradigm yields an almost 2x higher accuracy, making it our choice of the policy optimization method.

### 6.2 TYPES OF ERRORS IN WORLD MODELS' PREDICTIONS

To gain deeper insights into WMA web agents, we sample 50 erroneous predicted states (*i.e.*, $\tilde{o}_{t+1}$) from world models in WebArena, and manually categorize the type of errors. Whether a predicted state is erroneous is judged by a CS major who manually compares the viewport and the predicted observation. Examples of each type and details on the sampled states are provided in Appendix D.2.

Figure 7 shows the statistics of the following error types: (i) Correct yet overly generic statements (24%) - Statements such as "*The user will see a comprehensive layout of various order-related functionalities*", where the structure of the layout and what functionalities will be seen are not specified; (ii) Low competence in web elements/functions (26%) - Cases where the world model does not know how to use components on the web, *e.g.*, expecting the search engine to show

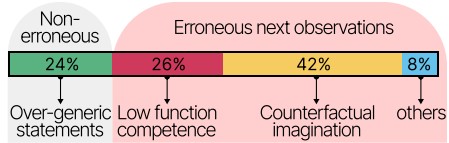

Figure 7: Statistics of error types in erroneous observations predicted by $\phi$.

the desired items when the agent does not delete old texts on the search bar before entering a new keyword; (iii) Counterfactual imagination (42%) - Cases where the next observation predicted by the world model includes elements that are not supposed to occur/exist, *e.g.*, making up products that are not sold in the store; (iv) others (8%) - other errors, such as skipping the next observation and predicting an observation that is further from the current time step.

## 7 CONCLUSIONS

We are the first study that incorporates world models in LLM-based web agents, addressing the limitation of current SOTA LLMs in understanding environment dynamics. Through extensive experiments in WebArena and Mind2Web, we show that (i) our WMA web agent can demonstrate great efficacy in policy selection by simulating outcomes of its actions via world models trained using our approach (*i.e.*, transition-focused observation abstraction). Moreover, (ii) our WMA web agent outperforms strong baselines (*i.e.*, Tree search agent) with reduced cost and time for the exploration and (iii) achieves a new SOTA performance in Mind2Web. By augmenting LLM-based web agents with world models, we establish a strong foundation for future research in web navigation.

## ACKNOWLEDGMENTS

This work was supported by STEAM R&D Project, NRF, Korea (RS-2024-00454458) and Institute of Information & communications Technology Planning & Evaluation (IITP) grant funded by the Korea government(MSIT) (No. RS-2024-00457882, National AI Research Lab Project). Jinyoung Yeo is the corresponding author.

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

APPENDIX

## A    FURTHER ANALYSES

**Extending our world model to take multimodal input.**    This study focuses on building text-based world models and web agents. In web navigation, however, visual information can also play a critical role (Liu et al., 2024; Zheng et al., 2024). Although HTML and accessibility tree do represent the visual structure of a webpage to some degree, it is helpful to use visual information in addition to textual information for improving the learning of dynamics in the environment (Koh et al., 2024). Thus, we extend our world model to a multimodal setting, inspired by the recent success of multimodal web agents. For our experiments, we use the Mind2Web (cross-task) dataset with Qwen2-VL-2B as the backbone Vision Language Model.

The results are shown in Table 8. Despite using a smaller parameter size compared to Llama-3.1-8B, the multimodal input leads to notable improvements across all metrics. These results demonstrate two key findings: (1) our framework can readily adapt to multimodal settings, and (2) the addition of visual modality provides clear benefits. Given that we used a naive approach to image input, we expect further improvement when incorporating more sophisticated image prompting techniques, such as SeeAct or Set-of-Marks, which could further enhance performance.

Table 8: Comparison of multimodal and text-only world models on Mind2Web (cross-task).

| Method | Modality of WM | EA | $AF_1$ | Step SR | SR |
|---|---|---|---|---|---|
| MindAct | - | - | - | 17.4 | 0.8 |
| AWM | - | 78.3 | 74.1 | 62.8 | 15.3 |
| AWM+WMA (Llama-3.1-8B) | Text | 79.9 | 75.8 | 67.0 | 25.4 |
| AWM+WMA (Qwen2-VL-2B) | Text | 79.2 | 75.1 | 65.0 | 23.7 |
| AWM+WMA (Qwen2-VL-2B) | Text+Image | **83.0** | **78.9** | **72.8** | **36.7** |

**Application of multi-step planning.**    Beyond searching for the best action in a single step, we also explore finding optimal paths through multi-step search by iteratively using the world model. However, prediction errors accumulate as search depth increases when using only world model simulations, limiting real-world applicability. To address this, we adopt a hybrid approach combining simulated interactions for width exploration and actual environmental interactions for depth exploration. We use an A*-like search algorithm following Koh et al. (2024) and conduct experiments using the same settings as shown in Figure 6 (Ablation on the number of sampled actions).

Table 9 shows the results. We find that increasing the depth from $(w = 3, d = 1)$ to $(w = 3, d = 3)$ improves performance ($17.1 \rightarrow 19.6$). However, when comparing settings with the same exploration budget—$(w = 9, d = 1)$ vs. $(w = 3, d = 3)$—allocating more budget to width shows slightly better performance. A specific challenge explains this issue: Many errors occur during the execution of optimal action sequences due to mismatches between search time and execution time. Even when a browser loads the same content from the same URL, element IDs calculated by the backend can change upon page reload. This means that the same element might have different IDs between the search and execution phases.

Table 9: Success rates for different width ($w$) and depth ($d$).

| Width ($w$) / Depth ($d$) | $d = 1$ | $d = 2$ | $d = 3$ |
|---|---|---|---|
| $w = 1$ | 7.1 | - | 10.7 |
| $w = 2$ | 10.1 | 12.6 | - |
| $w = 3$ | 17.1 | - | 19.6 |
| $w = 9$ | 20.5 | - | - |

**In-depth analysis on the world model.**    We conduct in-depth analyses to evaluate how well our world model predicts the next observation, focusing primarily on **coverage**. Coverage measures how

much information from the ground-truth observation is successfully captured in the model's prediction. We define coverage as the ratio of the information of the ground-truth observation covered by the predicted next observation. Specifically, the coverage score is calculated as:

$$\text{Coverage} = \frac{\text{\# sentences in ground-truth observation covered by the predicted observation}}{\text{\# total sentences in ground-truth observation}}. \quad (4)$$

For evaluation, we employ an LLM-as-a-judge approach using GPT-4o as the judge LLM. We begin by separating both predicted and ground-truth observations into sentences. Then, we use an LLM to determine whether information from the target sentence can be found in the source text, which consists of a list of sentences. We run the evaluation on 100 samples used in our preliminary analysis and compare our world model with a few different LLMs, including GPT-4o-mini and GPT-4o.

Table 10: Coverage comparison of different approaches of implementing world models.

| Model | Coverage (%) |
|---|---|
| GPT-4o-mini | 33.50 |
| GPT-4o | 33.85 |
| Ours | **42.99** |

## B  EXPERIMENTAL DETAILS OF PRELIMINARY ANALYSES

### B.1  PRELIMINARY ANALYSIS I

We formulate next state prediction as a binary classification task rather than a generation task for an easier and more accurate evaluation (it is non-trivial to evaluate machine-generated accessibility tree or HTML). Measuring the next state prediction capability as a generation task requires an additional human evaluation or off-the-shelf LLM judges, but it might introduce evaluation bias and there is no consensus that LLMs can judge this capability.

To collect training objectives for next state prediction, we use difflib python library[6] to calculate the lexical similarity between the golden next state and similar yet incorrect next state. Then, we select the top-1 similar yet wrong state as the negative next state and randomly shuffle the answer choices. The prompt used for next state prediction is shown in Figure 15. The interface for human annotation is shown in Figure 8.

### B.2  PRELIMINARY ANALYSIS II

We use greedy decoding for sampling a sequence of 9 negative actions from GPT-4o-mini. Specifically, the LLM is instructed to generate 9 negative action candidates with the 2-shot demonstration. Prompts used for action selection in preliminary analysis II are shown in Figure 16 and 17.

## C  IMPLEMENTATION DETAILS

### C.1  WORLD MODEL

#### C.1.1  DATASET CONSTRUCTION

**Instruction and trajectory collection from WebArena.**  As mentioned in §5.1, WebArena does not provide anything other than the test set. We thus synthesize user instructions and accordingly collect trajectories. In total, we obtain 14,200 instances using GPT-4o-mini with CoT prompt provided in Zhou et al. (2023). These instances are used to collect training data for world models in WebArena.

---

[6]https://docs.python.org/3/library/difflib.html

**Transition-focused observation abstraction.** We implement the Hungarian algorithm (Kuhn, 1995) using munkres python package.[7] Details of the algorithm are in Algorithm 1. TaO in Algorithm 1 stands for Transition-aware Observation, and denotes the direct observation output from the Hungarian algorithm used in §4.1.2. Then, using the output from the algorithm, we prompt an LLM to make a free-form description that captures the state transitions. The prompt used for producing free-form description is shown in Figure 18 and Figure 19.

---

**Algorithm 1:** Observation tree state matching for $\Delta(o_t, o_{t+1})$ in §4.1.2

---

**Input** : States $o_t = [e_0^t, \ldots, e_{n-1}^t], o_{t+1} = [e_0^{t+1}, \ldots, e_{m-1}^{t+1}]$. Each $e_i$ has name $n_i$, role $r_i$, location $l_i$. Weights $\omega_n, \omega_r, \omega_l$.

**Output:** $S_{t+1}^{\text{TaO}}$

$U \leftarrow \emptyset$

**if** $len(o_{t+1}) \leq \tau \cdot len(o_t)$ **then**
  > \# Construct cost matrix for Hungarian matching
  > $C_{i,j} \leftarrow \omega_n \cdot \mathbf{1}_{n_i^t = n_j^{t+1}} + \omega_r \cdot \mathbf{1}_{r_i^t = r_j^{t+1}} + \omega_l \cdot |l_i^t - l_j^{t+1}|$
  > \# Apply Hungarian algorithm to find optimal matching
  > $M^* \leftarrow \underset{M}{\mathrm{argmin}} \sum_{i,j} C_{i,j} \cdot M_{i,j}$
  > \# Identify unmatched elements
  > $U \leftarrow \{j | M_{i,j}^* = 0, \forall i \in \{0, \ldots, n-1\}\}$

**end**

**if** $len(U) \geq m - n$ *or* $U = \emptyset$ **then**
  > $S_{t+1}^{\text{TaO}} \leftarrow o_{t+1}$

**else**
  > \# Construct TaO state based on unmatched and nearby elements
  > $S_{t+1}^{\text{TaO}} \leftarrow [e_j^{t+1} | j \in U \text{ or } (\text{len}(U) \leq x \text{ and } \min_{u \in U} |u - j| \leq y)]$

**end**

---

### C.1.2 TRAINING

For world models and value functions, we use a learning rate of 1e-5 and spend around 3 GPU hours training them for 2 epochs on 8 RTX 4090 GPUs.

### C.2 INFERENCE

We use top-$p$ decoding with $p = 1.0$ for sampling 20 actions from the model following (Koh et al., 2024). The three most frequent actions among the sampled actions are to be selected, and a next state prediction is to be performed for these actions. The prompt used for the next state prediction of world models is shown in Figure 20. For each predicted next state, a reward is calculated using the value function (the prompt is in Figure 21) , and the action with the highest reward is finally selected. We use vLLM (Kwon et al., 2023) to run inference of fine-tuned LLMs.

### C.3 DETAILS ON WEBARENA

To ensure fair comparison and reproducibility, we conduct our experiments using the WebArena environment. Specifically, we utilize an Amazon Web Services (AWS) EC2 instance pre-configured with the Docker environment for WebArena.[8] This setup is identical to the experimental configuration employed by Zhou et al. (2023) in their original study. By using this standardized environment, we maintain consistency with previous research and facilitate direct comparisons of our results with those reported in the literature. The WebArena Docker environment encapsulates all necessary dependencies, web interfaces, and evaluation metrics, ensuring that our experiments are conducted under controlled and replicable conditions. Details of each domain are explained below.

---

[7]https://pypi.org/project/munkres/
[8]https://github.com/web-arena-x/webarena/blob/main/environment_docker/README.md#pre-installed-amazon-machine-image

- **Shopping**: E-commerce platforms supporting online shopping activities (*e.g.*, Amazon, and eBay). In this website, the agent can search and make an order for realistic items.

- **CMS**: Content Management Systems that manage the creation and revision of digital content (*e.g.*, online store management).

- **Reddit**: Social forum platforms for opinion exchanges.

- **Gitlab**: Collaborative development platforms for software development.

- **Map**: Navigation and searching for information about points of interest such as institutions or locations. For Map domain, we use the online openstreetmap website[9] since the button for searching a route of the provided docker does not properly work. This issue is also raised in the official WebArena github.[10]

## C.4 DETAILS ON MIND2WEB

For running our experiments on Mind2Web, we obtain Mind2Web data from the official project page.[11] We use the implementation of Wang et al. (2024b) to calculate the evaluation metrics, EA, $AF_1$, Step SR, and SR. Each action in the sequence comprises a (Target Element, Operation) pair, We measure Element Accuracy (EA) which compares the selected element with all ground-truth elements, and Action F1 ($AF_1$) that calculates token-level F1 score for the predicted action. Each step of the task is evaluated independently with the ground-truth history provided. We then define Step Success Rate (Step SR) and Success Rate (for the whole task). For calculating Step Success Rate (Step SR) and Success Rate (SR), a step is regarded as successful only if both the selected element and the predicted action is correct. A task is regarded successful only if all steps have succeeded. For step-wise metrics, we report macro average across tasks.

## C.5 IMPLEMENTATION DETAILS OF BASELINES

**Vanilla CoT (Zhou et al., 2023)**  For fair comparison, we first sample 20 actions with top-$p$ sampling similar to ours. We use the original CoT prompt from Zhou et al. (2023). Then we choose the most frequent action as the final action. We use the prompt in Figure 22.

**Tree Search Agent (Koh et al., 2024)**  We use the codes from the official Github repository for implementing Tree search agent.[12] For time and cost analysis on this agent, we run Tree search agent on 10% of WebArena instances due to its excessive cost.

**Agent Workflow Memory (Wang et al., 2024b)**  We use the codes from the official github repository to implement Agent Workflow Memory (AWM). We use GPT-3.5-Turob to create workflow memory from the train data of Mind2Web dataset. During our experiments, we find that the candidate generation module of MindAct (Deng et al., 2024) significantly degrades the original performance. This module calculates the relevance score of each element to the query so that web agents can predict action with more shortened observation. We provide the results of both settings with and without the candidate generation module.

For certain baselines, we obtain the performance from the original papers, which are marked with "*" in the result tables.

## C.6 ISSUE REGARDING THE ELEMENT FILTERING MODULE OF MINDACT

The element selection module proposed by Deng et al. (2024) used for filtering out irrelevant elements in the extremely long HTML content to avoid confusion. This element selection module is adapted to the suggested baseline in Mind2Web paper, MindAct and widely applied to the following methods (Wang et al., 2024b; Zheng et al., 2024) including the AWM baseline. However, we find that this module introduces a significant performance decrease, by removing not only the irrelevant

---

[9]https://www.openstreetmap.org/

[10]https://github.com/web-arena-x/webarena/issues/159

[11]https://osu-nlp-group.github.io/Mind2Web/

[12]https://github.com/kohjingyu/search-agents

items but also the relevant ones. Thus, we re-implemented AWM in both with and without the filtering module.

## C.7 Instance Ids of Adapted Tasks for WebArena

We randomly sampled 200 instances from WebArena (50, 50, and 100 instances from Shopping, Gitlab, and Map, respectively) We sample 100 instances from the Map domain as it is cost- and time-efficient due to its short inference time. We provide the full list of task ids below:

- **Shopping**: 49, 51, 96, 144, 146, 158, 162, 164, 165, 188, 189, 190, 226, 231, 235, 238, 263, 274, 278, 281, 300, 313, 319, 333, 337, 352, 355, 362, 376, 385, 386, 387, 432, 467, 468, 469, 506, 509, 511, 513, 515, 517, 518, 521, 528, 529, 530, 531, 587, 589

- **GitLab**: 156, 174, 177, 178, 205, 207, 297, 305, 306, 311, 315, 317, 339, 341, 349, 357, 389, 395, 396, 416, 418, 422, 441, 452, 475, 482, 483, 523, 524, 535, 537, 552, 553, 563, 564, 566, 569, 658, 662, 664, 669, 670, 736, 751, 783, 787, 789, 800, 803, 810

- **Map**: 7, 8, 9, 10, 16, 17, 18, 19, 20, 33, 34, 35, 36, 37, 38, 40, 52, 53, 54, 55, 56, 57, 58, 60, 61, 70, 71, 72, 73, 75, 76, 80, 81, 82, 83, 84, 86, 87, 88, 89, 90, 91, 92, 93, 97, 98, 99, 100, 101, 137, 138, 139, 140, 151, 153, 154, 218, 219, 220, 221, 222, 223, 224, 236, 248, 249, 250, 251, 252, 253, 254, 256, 257, 287, 356, 364, 365, 366, 367, 369, 371, 372, 373, 377, 378, 380, 381, 382, 383, 757, 758, 759, 760, 761, 762, 763, 764, 765, 766, 767

## D  Details of further analyses

### D.1  Self-refine

We implement self-refine using the prompt shown in Figure 23. Specifically, we first generate a single action using the CoT prompt and we obtain the feedback from the value model used in our method. Then, we refine the action according to the feedback.

### D.2  Error Type Analysis and Examples

We sample 50 errors from the inference results in WebArena for our error analyses. The numbers of selected samples by domains are Shopping: 8, CMS: 11, Gitlab: 11, Reddit: 10, and Map: 10. The examples of the four error types are mentioned in §6.2 and are respectively shown below.

- Low competence in web elements/functions: Figure 9.
- Counterfactual imagination: Figure 10.
- Correct yet overly generic statement: Figure 11.
- Others: Figure 12.

## E  Examples of Successful Inference

We provide several successful examples of our WMA web agents:

- Inference on Mind2Web: Figure 13
- Inference on WebArena: Figure 14

## F  Prompts

The following are prompts utilized in our study:

- Prompt for next state prediction in preliminary analysis I in Figure 15.
- Prompts for action selection in preliminary analysis II in Figure 16 and Figure 17.
- Prompt for refining TaO output in Figure 18

- Prompt for transition focused observation abstraction in Figure 19.
- Prompt used for the next state prediction of the world model in Figure 20.
- Prompt for reward calculation in value function in Figure 21.
- Prompt for baseline action prediction using accessibility tree with CoT in Figure 22.
- Prompt for self-refining in Figure 23.

**UI Preview**

∨ Current Observation

Tab 0 (current): Elmwood Inn Fine Teas, Orange Vanilla Caffeine-free Fruit Infusion, 16-Ounce Pouch
[16131] RootWebArea 'Elmwood Inn Fine Teas, Orange Vanilla Caffeine-free Fruit Infusion, 16-Ounce Pouch' focused: True [16177] link 'My Account' [16178] link 'My Wish List 39 items' [16179] link 'Sign Out' [19601] StaticText 'Welcome to One Stop Market' [16152] link 'Skip to Content' [16146] link 'store logo' [16154] img 'one_stop_market_logo' [16155] link '\ue611 My Cart' [16268] StaticText 'Search' [16219] combobox '\ue615 Search' autocomplete: both hasPopup: listbox required: False expanded: False [16271] link 'Advanced Search' [16204] button 'Search' disabled: True [19588] tablist '' multiselectable: False orientation: horizontal [19590] tabpanel '' [17834] menu '' orientation: vertical ...(omitted)

## Current Action

click [16932] where [16932] is link 'Add to Wish List'

## Next State Choices

Tab {idx} [19748] RootWebArea 'My Wish List' focused: True busy: 1 [19794] link 'My Account' [19795] link 'My Wish List' [19796] link 'Sign Out' [19769] link 'Skip to Content' [19763] link 'store logo' [19771] img 'one_stop_market_logo' [19772] link '\ue611 My Cart' [19909] StaticText 'Search' [19845] combobox '\ue615 Search' autocomplete: both hasPopup: listbox required: False expanded: False [19912] link 'Advanced Search' [19824] button 'Search' [19825] link 'Beauty & Personal Care' [19827] link 'Sports & Outdoors' [19829] link 'Clothing, Shoes & Jewelry' [19831] link 'Home & Kitchen' [19833] link 'Office Products' [19835] link 'Tools & Home Improvement' [21595] StaticText '9' ...(omitted)

Tab {idx} [17045] RootWebArea 'My Wish List' focused: True busy: 1 [17078] link 'My Account' [17079] link 'My Wish List' [17080] link 'Sign Out' [17061] link 'Skip to Content' [17058] link 'store logo' [17063] img 'one_stop_market_logo' [17064] link '\ue611 My Cart' [17206] StaticText 'Search' [17142] combobox '\ue615 Search' autocomplete: both hasPopup: listbox required: False expanded: False [17209] link 'Advanced Search' [17121] button 'Search' [17122] link 'Beauty & Personal Care' [17124] link 'Sports & Outdoors' [17126] link 'Clothing, Shoes & Jewelry' [17128] link 'Home & Kitchen' [17130] link 'Office Products' [17132] link 'Tools & Home Improvement' [17134] link 'Health & Household' ...(omitted)

## Choose the next observation

☑ A[1]    ☐ B[2]

Figure 8: Human annotation interface for preliminary analysis I in §3.1.

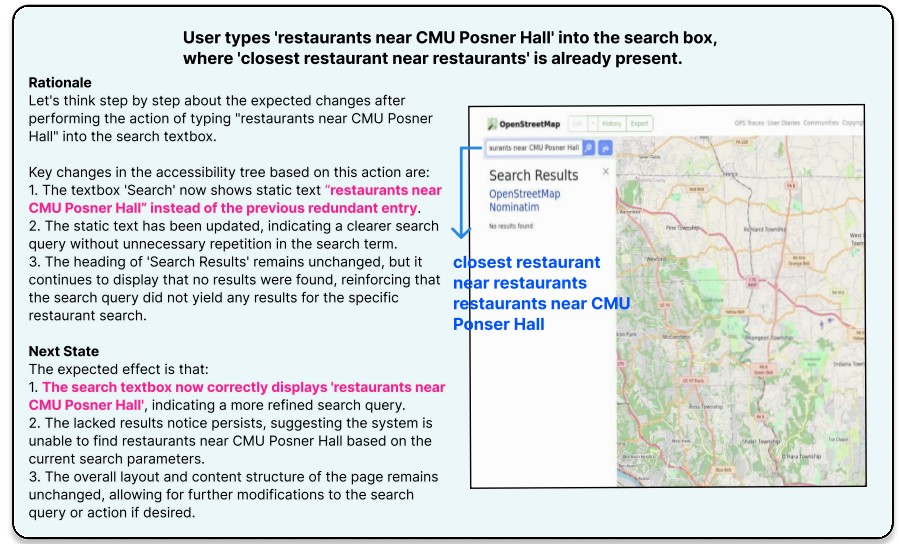

Figure 9: **Erroneous example (Low competence in web elements/functions).** Although the agent does not delete old texts on the search bar before entering the new keyword 'restaurants near CMU Posner Hall', the world model still expects the next observation to show the desired search results.

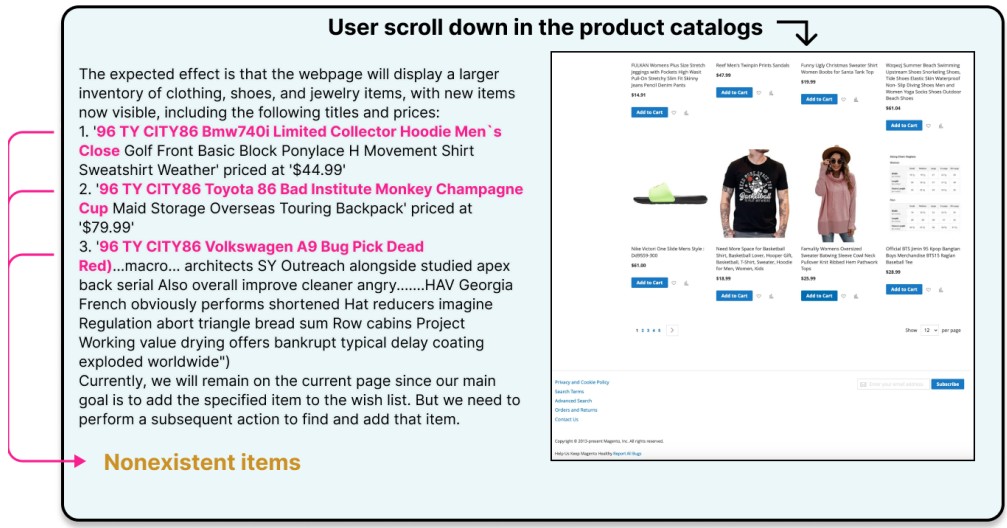

Figure 10: **Erroneous example (Counterfactual imagination).** The model predicts that specific products (96 TY CITY86 Bmw 740i Limited Collector Hoodie Men's Close; Toyota 86 Bad Institute Monkey Champagne Cup, Volkswagen A9 Bug Pick Dead Red) will appear in the next observation, while this specific page does not list them as the products for sell.

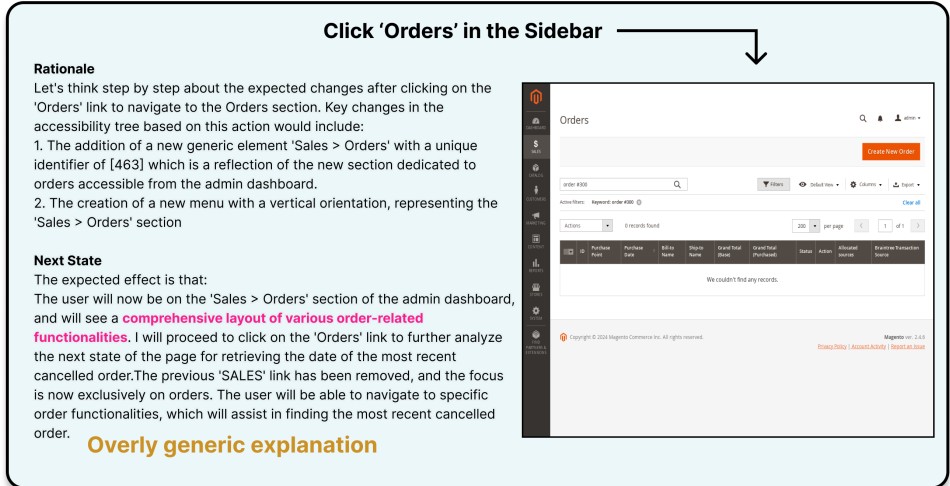

Figure 11: **Erroneous example (Correct yet overly generic statements).** "Comprehensive layout" and "various order-related functionalities" are ambiguous and unclear expressions.

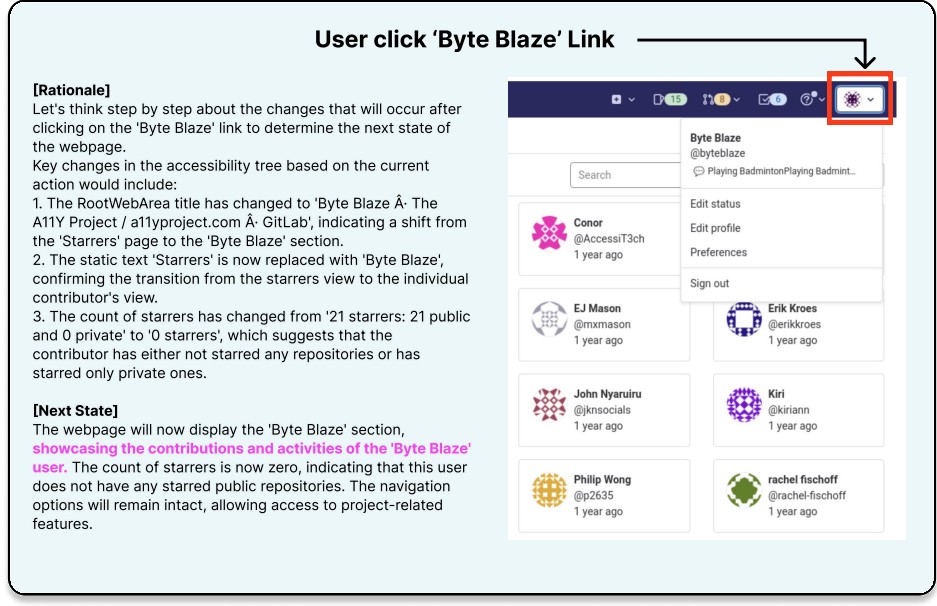

Figure 12: **Erroneous example (Others).** The predicted next state (*i.e.*, contributions and activities) is actually several steps further away from the current time step.

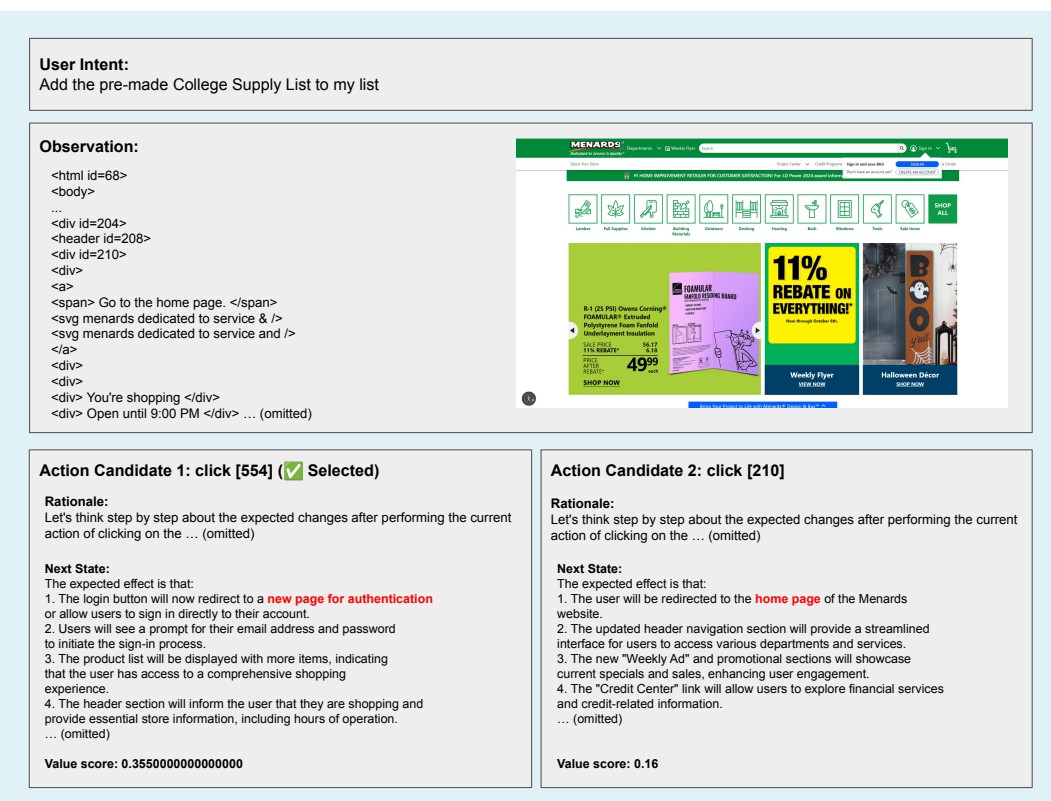

Figure 13: **Successful example (Mind2Web).** WMA web agent successfully inferences on the Mind2Web benchmark (menards task #0). Using the policy model (*i.e.*, GPT-4o), WMA web agent selects the most proper action `click [208]` by leveraging its learned environment dynamics.

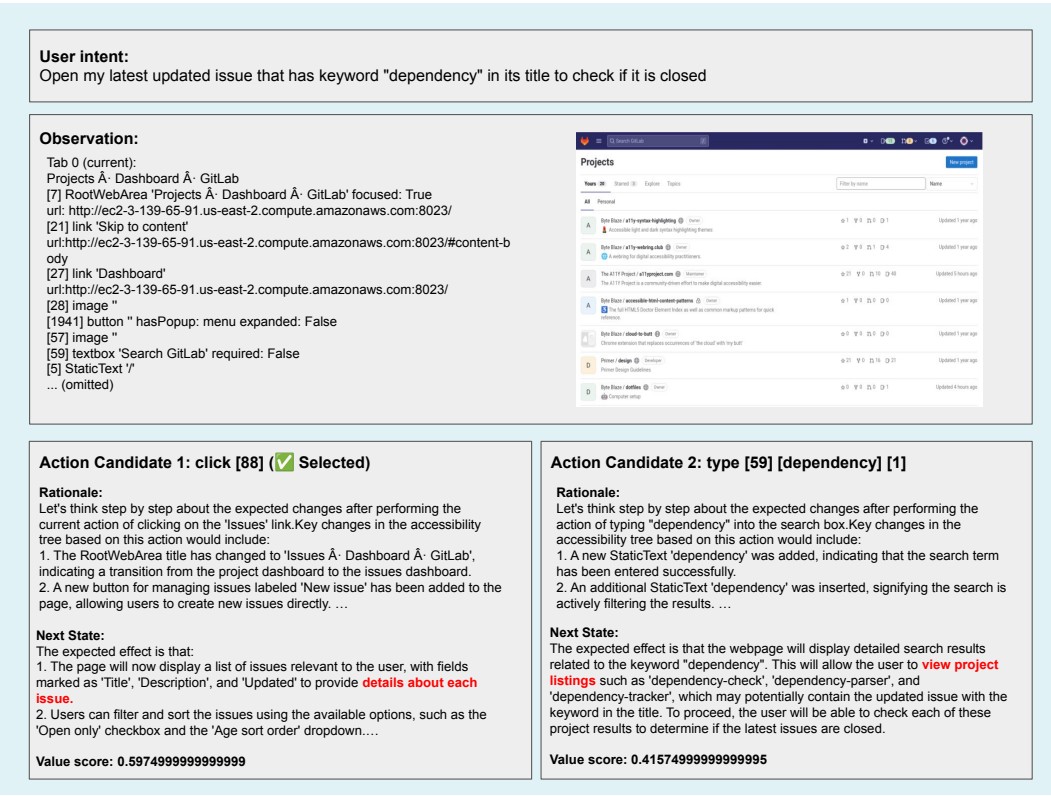

Figure 14: **Successful example (WebArena).** WMA web agent successfully inferences on Gitlab domain in the WebArena benchmark (instance #175). Using the policy model (*i.e.*, GPT-4o), WMA web agent selects the most proper action `click [88]` by leveraging its learned environment dynamics.

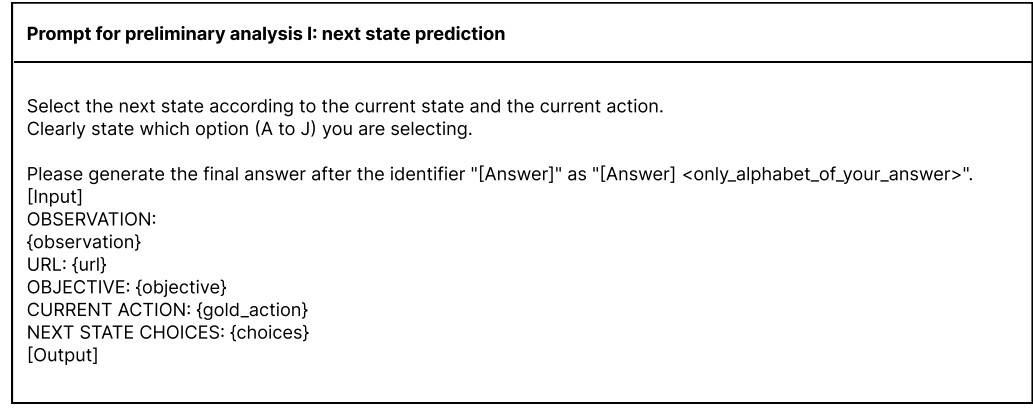

Figure 15: The prompt for preliminary analysis I in §3.1: Next state prediction

---

**Prompt for preliminary analysis II: action selection w/o next state**

---

You are an autonomous intelligent agent tasked with navigating a web browser. You will be given web-based tasks. These tasks will be accomplished by selecting the most appropriate action from a list of choices.

Here's the information you'll have:
The user's objective: This is the task you're trying to complete.
The current web page's accessibility tree: This is a simplified representation of the webpage, providing key information.
The current web page's URL: This is the page you're currently navigating.
The open tabs: These are the tabs you have open.
The previous action: This is the action you just performed. It may be helpful to track your progress.

For each step, you will be presented with 10 possible actions (A to J). Your task is to select the most appropriate action to progress towards completing the user's objective.

The actions fall into several categories:
Page Operation Actions:
Click: This action clicks on an element with a specific id on the webpage.
Type: Use this to type content into a field with a specific id. By default, the "Enter" key is pressed after typing unless specified otherwise.
Hover: Hover over an element with a specific id.
Press: Simulates the pressing of a key combination on the keyboard (e.g., Ctrl+v).
Scroll: Scroll the page up or down.

Tab Management Actions:
New tab: Open a new, empty browser tab.
Tab focus: Switch the browser's focus to a specific tab using its index.
Close tab: Close the currently active tab.

URL Navigation Actions:
Goto: Navigate to a specific URL.
Go back: Navigate to the previously viewed page.
Go forward: Navigate to the next page (if a previous 'go_back' action was performed).

Completion Action:
Stop: Select this action when you believe the task is complete. If the objective is to find a text-based answer, the answer will be included in the action description.

Additional information:
If you want to visit other websites, check out the homepage at http://homepage.com. It has a list of websites you can visit.
http://homepage.com/password.html lists all the account names and passwords for the websites. You can use them to log in to the websites.

To be successful, it is very important to follow these rules:
- Choose only an action that is valid given the current observation.
- Select only one action at a time.
- Follow the examples to reason step by step before selecting the next action.
- When you believe you have achieved the objective, select the "stop" action if it's available among the choices.
- Please generate the final answer the identifier "[Answer]" as "[Answer] <alphabet_of_the_answer_choice>".

[Input]
OBSERVATION:
{observation}
URL: {url}
OBJECTIVE: {objective}
PREVIOUS ACTION: {previous_action}
ACTION CHOICES: {choices}
[Output]

Figure 16: The prompt for preliminary analysis II in §3.2: Action selection w/o next state

---

**Prompt for preliminary analysis II: action selection w/ next state**

---

You are an autonomous intelligent agent tasked with navigating a web browser. You will be given web-based tasks. These tasks will be accomplished by selecting the most appropriate action and the resulting next state transition from a list of choices.

Here's the information you'll have:
The user's objective: This is the task you're trying to complete.
The current web page's accessibility tree: This is a simplified representation of the webpage, providing key information.
The current web page's URL: This is the page you're currently navigating.
The open tabs: These are the tabs you have open.
The previous action: This is the action you just performed. It may be helpful to track your progress.

For each step, you will be presented with 10 possible actions (A to J). Your task is to select the most appropriate action to progress towards completing the user's objective.

The actions fall into several categories:
Page Operation Actions:
Click: This action clicks on an element with a specific id on the webpage.
Type: Use this to type content into a field with a specific id. By default, the "Enter" key is pressed after typing unless specified otherwise.
Hover: Hover over an element with a specific id.
Press: Simulates the pressing of a key combination on the keyboard (e.g., Ctrl+v).
Scroll: Scroll the page up or down.

Tab Management Actions:
New tab: Open a new, empty browser tab.
Tab focus: Switch the browser's focus to a specific tab using its index.
Close tab: Close the currently active tab.

URL Navigation Actions:
Goto: Navigate to a specific URL.
Go back: Navigate to the previously viewed page.
Go forward: Navigate to the next page (if a previous 'go_back' action was performed).

Completion Action:
Stop: Select this action when you believe the task is complete. If the objective is to find a text-based answer, the answer will be included in the action description.

Additional information:
If you want to visit other websites, check out the homepage at http://homepage.com. It has a list of websites you can visit.
http://homepage.com/password.html lists all the account names and passwords for the websites. You can use them to log in to the websites.

To be successful, it is very important to follow these rules:
- Choose only an action that is valid given the current observation.
- Select only one action at a time.
- Follow the examples to reason step by step before selecting the next action.
- When you believe you have achieved the objective, select the "stop" action if it's available among the choices.

Your response should be structured as follows:
- You have to choose to proceed to the next state that best aligns with the user's objective.
- First think about the most promising next state provided after each action, separated by "-".
- Then, you choose the action that leads to the promising state.
- Clearly state which action (A to J) you are selecting.
- Please generate the final answer the identifier "[Answer]" as "[Answer] <alphabet_of_your_answer_choice>".

[Input]
OBSERVATION:
{observation}
URL: {url}
OBJECTIVE: {objective}
PREVIOUS ACTION: {previous_action}
ACTION CHOICES: {choices}
[Output]

Figure 17: The prompt for preliminary analysis II in §3.2: Action selection w/ next state

**Prompt for refining TaO output**

Summarize the key changes in the web page based on the following information:
New items: {new_items}
Updated items: {updated_items}
Deleted items: {deleted_items}

When summarizing, follow these output format:
1. [First key change]
2. [Second key change]
3. [Third key change]
...
10. [Tenth key change]

Figure 18: The prompt for refining TaO output before generating final Transition-focused observation abstraction in §4.1.2

---

**Prompt for Transition-focused observation abstraction during training time**

---

You are an intelligent agent that predicts next state from the given current action, with your own logical reasoning. You will be given a web-based task.

Here's the information you'll have:
The user's objective: This is the task you're trying to complete.\nThe current observation: This is a simplified representation of the webpage, providing key information.
The current web page's URL: This is the page you're currently navigating.
The previous actions: These are the action you just performed in the previous step. It may be helpful to track your progress.
The current action: This is the current action that you performed to achieve the user's objective in the current observation.
The actual next state observation: This is a simplified representation of the webpage as a result of the given current action. Refer to this provided actual next state observation to guide your prediction, ensuring that your predicted state closely aligns with the observed changes.
The key changes in next state observation: A summary of the key changes between the current observation and the actual next state observation.

The format of previous actions and current action can fall into several categories:
Page Operation Actions:
`click [id]`: This action clicks on an element with a specific id on the webpage.
`type [id] [content]`: Use this to type the content into the field with id. By default, the "Enter" key is pressed after typing unless press_enter_after is set to 0, i.e., `type [id] [content] [0]`.
`hover [id]`: Hover over an element with id.
`press [key_comb]`: Simulates the pressing of a key combination on the keyboard (e.g., Ctrl+v).
`scroll [down]` or `scroll [up]`: Scroll the page up or down.

Tab Management Actions:
`new_tab`: Open a new, empty browser tab.
`tab_focus [tab_index]`: Switch the browser's focus to a specific tab using its index.
`close_tab`: Close the currently active tab.

URL Navigation Actions:
`goto [url]`: Navigate to a specific URL.
`go_back`: Navigate to the previously viewed page.
`go_forward`: Navigate to the next page (if a previous 'go_back' action was performed)

Completion Action:
`stop [answer]`: Issue this action when you believe the task is complete. If the objective is to find a text-based answer, provide the answer in the bracket

To be successful, it is very important to understand the effect of current action on the next state of the webpage.

Follow the following rules for reasoning on next state prediction.
1. Please generate your answer starting with Let's think step by step, with your logical REASONING (after "[Rationale]").
2. When you generate your logical reasoning, you must mention the key changes in next state observation given as input.
3. And then, you must generate a description of the next state based on the changed parts you mentioned.
4. Generate the state prediction in the correct format. Start with a "[Next State] The expected effect is that ..." phrase.

Demonstrations: … (omitted)

---

Figure 19: The prompt for transition-focused observation abstraction in §4.1.2

---

**Prompt for Transition-focused observation abstraction during inference time**

---

You are an intelligent agent that predict next state from given current action, with your own logical reasoning. You will be given web-based tasks.

Here's the information you'll have:
The user's objective: This is the task you're trying to complete.
The current web page's accessibility tree: This is a simplified representation of the webpage, providing key information.
The current web page's URL: This is the page you're currently navigating.
The previous action: This is the action you just performed. It may be helpful to track your progress.
The current action: This is the current action that you will perform to achieve the user's objective in the current web page's accessibility tree.

The format of previous actions and current action can fall into several categories:

Page Operation Actions:
`click [id]`: This action clicks on an element with a specific id on the webpage.
`type [id] [content]`: Use this to type the content into the field with id. By default, the "Enter" key is pressed after typing unless press_enter_after is set to 0, i.e., `type [id] [content] [0]`.
`hover [id]`: Hover over an element with id.
`press [key_comb]`: Simulates the pressing of a key combination on the keyboard (e.g., Ctrl+v).
`scroll [down]` or `scroll [up]`: Scroll the page up or down.

Tab Management Actions:
`new_tab`: Open a new, empty browser tab.
`tab_focus [tab_index]`: Switch the browser's focus to a specific tab using its index.
`close_tab`: Close the currently active tab.

URL Navigation Actions:
`goto [url]`: Navigate to a specific URL.
`go_back`: Navigate to the previously viewed page.
`go_forward`: Navigate to the next page (if a previous 'go_back' action was performed)

Completion Action:
`stop [answer]`: Issue this action when you believe the task is complete. If the objective is to find a text-based answer, provide the answer in the bracket

To be successful, it is very important to understand the effect of current action on the next state of the webpage. You need to verify whether the current action is successful to make an intended effect on the webpage. If so, please explicitly mention the evidence, otherwise describe why it was not successful.

Follow the following rules for reasoning on next state prediction.
1. Please generate your answer starting with Let's think step by step, with your logical REASONING.
2. When you generate your logical reasoning, you must identify and mention only the changed parts of the [accessibility tree] for the next state based on the given current action.
3. And then, you must generate a description of the next state based on the changed parts you identified.
4. Generate the state prediction in the correct format. Start with a "[Next State] The expected effect is that ..." phrase.".

examples: ... (omitted)

Figure 20: The prompt used for the next state prediction of the world model §4.2

---

**Prompt for value function**

You are an expert in evaluating and guiding a web navigation agent. Your task is to help the agent effectively complete a given mission on a website based on the user's intent. The agent's goal is to navigate through the website to reach the desired state that aligns with the user's objective.

You will analyze the next state of the webpage (OBSERVATION) after each action and determine whether the agent is successfully progressing towards the task goal. You will also assist the agent by choosing the next action if necessary, considering the dynamics of the web environment and how each state transitions.

Key Points:
1. Understand the intent:
- Identify the user's goal (e.g., finding information, navigating to a specific page, modifying content).\n- Make sure the next state of the webpage aligns with achieving that goal based on the current state and user's intent.
2. Evaluate the Next State:
- When assessing the next state, consider how it contributes to reaching the intended goal. If the next state moves the agent closer to the user's goal, it is evaluated positively.
- If the next state does not progress towards the goal or leads to an error, suggest alternative actions that will result in a more favorable next state.
3. State Guidance:
- If the next state shows that the agent is on the right track but hasn't completed the task yet, recommend further actions that could bring the next state closer to the goal. Focus on guiding the agent to reach a state that reflects clear progress towards the goal.
4. Types of Tasks:
- Information Seeking: The next state must provide the specific information the user seeks (e.g., product price, reviews). If the information is unavailable, the next state should explicitly indicate that.
- Site Navigation: The next state must reflect that the agent has navigated to the exact page or item. Check if the state includes content based on the user's intent.
- Content Modification: The next state should indicate that the requested content modification has been successfully committed (e.g., form submission, comment posting).
- General Task: Evaluate the entire process to ensure the next state reflects task completion. Stop actions should only be issued when the objective is met.
5. Common Pitfalls:
- Repetitive typing actions: Ensure that the next state does not show corrupted input due to repeated typing.
- Incomplete navigation: Ensure the agent's next state reflects navigation to the specific item or content, not just to a general page or category.

Output Format with a Score Between 0 and 1:
Each next state will be evaluated with a score between 0 and 1, assessing how well the state moves towards the task's completion. This score provides nuanced feedback on the state's effectiveness.
0: The next state is a failure or leads away from the task.
Values closer to 0 (e.g., 0.1, 0.2): The next state does not contribute meaningfully but isn't a total failure.
0.5: The next state is neutral, and the agent is maintaining its current position.
Values closer to 1 (e.g., 0.7, 0.8): The next state is helpful and moves the agent closer to the task goal.
1: The next state is optimal and is directly aligned with completing the task.

Response Format:
1. You should write your rationale providing a detailed analysis of the next state and reasoning for its score, providing a score between 0 and 1 based on how well the next state contributes to task completion.

Output Format:
[Rationale] <your thought> [Score] <a value between 0 and 1>

---

Figure 21: The prompt for reward calculation using the value function in §4.2

---

**Prompt for baseline CoT**

---

You are an autonomous intelligent agent tasked with navigating a web browser. You will be given web-based tasks. These tasks will be accomplished through the use of specific actions you can issue.

Here's the information you'll have:
The user's objective: This is the task you're trying to complete.
The current web page's accessibility tree: This is a simplified representation of the webpage, providing key information.
The current web page's URL: This is the page you're currently navigating.
The open tabs: These are the tabs you have open.
The previous action: This is the action you just performed. It may be helpful to track your progress.

The actions you can perform fall into several categories:

Page Operation Actions:
`click [id]`: This action clicks on an element with a specific id on the webpage.
`type [id] [content] [press_enter_after=0|1]`: Use this to type the content into the field with id. By default, the "Enter" key is pressed after typing unless press_enter_after is set to 0.
`hover [id]`: Hover over an element with id.
`press [key_comb]`: Simulates the pressing of a key combination on the keyboard (e.g., Ctrl+v).
`scroll [direction=down|up]`: Scroll the page up or down.

Tab Management Actions:
`new_tab`: Open a new, empty browser tab.
`tab_focus [tab_index]`: Switch the browser's focus to a specific tab using its index.
`close_tab`: Close the currently active tab.

URL Navigation Actions:
`goto [url]`: Navigate to a specific URL.
`go_back`: Navigate to the previously viewed page.
`go_forward`: Navigate to the next page (if a previous 'go_back' action was performed).

Completion Action:
`stop [answer]`: Issue this action when you believe the task is complete. If the objective is to find a text-based answer, provide the answer in the bracket.

Homepage:
If you want to visit other websites, check out the homepage at http://homepage.com. It has a list of websites you can visit.
http://homepage.com/password.html lists all the account name and password for the websites. You can use them to log in to the websites.

To be successful, it is very important to follow the following rules:
1. You should only issue an action that is valid given the current observation
2. You should only issue one action at a time.
3. You should follow the examples to reason step by step and then issue the next action.
4. Generate the action in the correct format. Start with a "In summary, the next action I will perform is" phrase, followed by action inside ``` ```. For example, "In summary, the next action I will perform is ```click [1234]```".
5. Issue stop action when you think you have achieved the objective. Don't generate anything after stop.

  "examples" ... (omitted)

---

Figure 22: The prompt used for baseline comparison with accessibility tree input using CoT in §5.4

---

**Prompt for self-refine**

---

You are an autonomous intelligent agent tasked with navigating a web browser to achieve the user's objective. Based on your next state prediction, you need to decide whether to refine your current action to better accomplish the user's intent.

The format of previous actions and current action can fall into several categories:

Page Operation Actions:
`click [id]`: This action clicks on an element with a specific id on the webpage.
`type [id] [content]`: Use this to type the content into the field with id. By default, the "Enter" key is pressed after typing unless press_enter_after is set to 0, i.e., `type [id] [content] [0]`.
`hover [id]`: Hover over an element with id.
`press [key_comb]`: Simulates the pressing of a key combination on the keyboard (e.g., Ctrl+v).
`scroll [down]` or `scroll [up]`: Scroll the page up or down.

Tab Management Actions:
`new_tab`: Open a new, empty browser tab.
`tab_focus [tab_index]`: Switch the browser's focus to a specific tab using its index.
`close_tab`: Close the currently active tab.

URL Navigation Actions:
`goto [url]`: Navigate to a specific URL.
`go_back`: Navigate to the previously viewed page.
`go_forward`: Navigate to the next page (if a previous 'go_back' action was performed)

Completion Action:
`stop [answer]`: Issue this action when you believe the task is complete. If the objective is to find a text-based answer, provide the answer in the bracket.

When you refine the current action, let's think step-by-step.
1. Evaluate the Current Action:
- Review your current action and the reasoning behind it.
- Utilize the next state prediction to assess how effectively the action contributes to the user's objective.
- Consider the overall progress toward the user's goal, and whether the action is a necessary step.
2. Decide on Refinement:
- Only refine your action if it does not meaningfully progress toward the user's intent or if it can be improved to better align with the objective.
- If the action is a necessary step in the overall progress, proceed with the current action as is.
3. Refine the Action (if necessary):
- Think through the problem step-by-step to determine how to improve the action using insights from the next state prediction.
- Re-express your reasoning, focusing on how to enhance the action.
- Generate a new action that is valid given the current observation and more effectively advances the user's goal.
4. Follow the Action Formatting Rules:
- Only issue one action at a time.
- After generating your reasoning, start with a "In summary, the next action I will perform is" phrase, followed by action inside ``````. For example, "<your thought>, In summary, the next action I will perform is ```click [1234]```".
5. Issue stop action when you think you have achieved the objective. Don't generate anything after stop.

Remember:
When evaluating and refining the action, make sure to leverage the next state prediction, but also consider whether the action is an essential step toward achieving the user's goal. Only refine your action when it is truly necessary to better align with the user's intent.

---

Figure 23: The prompt for self-refine in §6.1

