# OpenReview forum: "Web Agents with World Models: Learning and Leveraging Environment Dynamics in Web Navigation"
_ICLR.cc/2025/Conference — ICLR 2025 Poster_

### Official Review · Reviewer_tKJR · 2024-11-01

**Soundness:** 3
**Presentation:** 3
**Contribution:** 2
**Rating:** 6
**Confidence:** 4

**Summary:**

This paper explores the introduction of world knowledge models into the field of agents, particularly within the context of web navigation. Specifically, the study finds through exploratory experiments that
1. existing large models lack sufficient predictive capability regarding the consequences of their actions.
2. Enabling these models to anticipate such consequences can significantly enhance their planning capabilities.
Based on this finding, the authors trained a world knowledge model (WKM) to predict the potential outcomes of an action.  The proposed transition-focused observation abstraction plays a crucial role in the training stage.
The paper conducts comprehensive experiments to demonstrate the necessity and superiority of using WKM.

**Strengths:**

1. The introduction of this paper is well-constructed. Through experiments, it identifies the issue that existing models cannot effectively predict the potential consequences of their actions, and it highlights that enabling models to understand these consequences can enhance their planning capabilities. The proposed approach is very reasonable.
2. The explanations in this paper are clear and easy to understand. The formulas and figures effectively illustrate the method proposed by the authors.
3. The authors conducted multi-dimensional experiments that effectively address potential questions from readers.

**Weaknesses:**

1. Although this paper claims to be the first to introduce a world knowledge model into LLM-based agents, to my knowledge, this may not be the case. For example, the work "Agent Planning with World Knowledge Model" might also explore this area. The authors may need to conduct a more extensive literature review to compare and discuss the similarities and differences between their proposed method and existing approaches.
2. The performance improvement brought by the experiments in this paper is not significant. Additionally, the authors' WKM is limited to a depth of 1. It might be worth considering deeper levels.

**Questions:**

1. As discussed in weakness 1, could the authors conduct a more extensive literature review to discuss the similarities and differences between existing WKM-based methods and their proposed approach? Although this is mentioned in section 6.1, too few works are considered.
2. As discussed in weakness 2, while the authors have considered the impact of exploration breadth in WKM during the ablation study, could they conduct experiments to consider the impact of depth as well?
If the authors can adequately address the questions above, I would be willing to reconsider the review.

---

> ### Author Response · Authors · 2024-11-26
> **Response to Reviewer tKJR**
>
> We greatly appreciate your valuable feedback for improving our work.
>
> ### **Q1. A more detailed literature review of the related work**
> We agree that “Agent Planning with World Knowledge Model (WKM)” is highly related work and we elaborate more on the similarities and the differences between ours and WKM below.
>
> ### **Differences**
>
>
> **WKM predicts the task knowledge required to generate actions not the effect of actions:** WKM aims to provide knowledge for understanding the current state (i.e., state knowledge) and high-level guidelines for planning (i.e., task knowledge) prior to action prediction, such as "*The egg is most likely in the fridge ... The workflows are: 1) locate and take the egg; 2) clean the egg using sinkbasin ..*". Thus, the task knowledge $\kappa$ is predicted by the world knowledge model $\phi$, solely based on the user instruction $I$, i.e., $\pi_{\phi}(\kappa|I)$. Then, the policy agent $\theta$ predicts a trajectory $\tau$ based on the task knowledge, i.e., $\pi_{\theta}( \tau | \kappa, I)$.
>
>
> Our world model, however, focuses on predicting **the effect of specific actions on the environment** rather than the prior knowledge before the task. This approach aligns more closely with the definition of a world model, expressed as $p(s_{t+1}|s_{t}, a_{t})$, which predicts the next state given the current state and action.
>
> While WKM and our world model are designed for different purposes, we believe these models can be used simultaneously to provide agents with better understanding of tasks and current states, while helping to mitigate negative impacts on the environment.
>
> **WKM needs expert annotation:** WKM requires expert trajectories for training the knowledge model, which is difficult to obtain in real-life scenarios. In contrast, our world model is designed to learn environment dynamics solely from agent trajectories that can be automatically generated without expert annotations.
>
>
> ### **Similarities**
>
> **Both ours and WKM learn from the environment and guide selecting proper actions:**
> The key similarities between WMA web agents and WKM are leveraging trajectories to learn information from the environment and guiding the agents with it. However, as we mentioned above, our work and WKM focus on different aspects of information.
>
>
> ### **Other works related to WKM**
>
> A line of work has explored leveraging the knowledge that can obtained from the experience within the environment to learn from success and failure, similar to WKM:
> - AutoGuide [1]: To leverage the previous experience on the environment to improve LLM-based agents, they first extract knowledge (guideline) that helps predict a proper action and retrieve it when the agent faces similar situations. The extracted knowledge is similar to the task knowledge of the WKM paper.
>
> - KnowAgent [2]: KnowAgent extracts action knowledge, which consists of a sequence of actions and the description on it, and saves the knowledge into a knowledge base to incorporate the knowledge for planning at inference time.
>
> We will include a comprehensive discussion of this research direction in the related work section of our paper.
>
> [1] AutoGuide: Automated Generation and Selection of State-Aware Guidelines for Large Language Model Agents
>
> https://arxiv.org/pdf/2403.08978v1
>
> [2] KnowAgent: Knowledge-Augmented Planning for LLM-Based Agents
>
> https://arxiv.org/pdf/2403.03101
>
> ### **Q2. Impact of depth in exploration**
> We appreciate the suggestion of an interesting experiment. We additionally conduct the experiments and provide the results in the general response.

---

> > ### Comment · Reviewer_tKJR · 2024-11-27
> >
> > Thank you for your thorough and thoughtful response.
> > I am pleased to see that you have clearly articulated the similarities and differences between your work and existing studies in the related work section. This has helped clarify the contributions of your work.
> > I am also glad to see that you explored the impact of different depths on your experimental results. Conducting such experiments in a short amount of time is quite complex, and I look forward to seeing a more detailed discussion on this aspect in the revised version.
> > Overall, your response has effectively addressed my concerns. I will be adjusting my score to a 6. Good luck with your submission!

---

> > > ### Author Response · Authors · 2024-11-27
> > >
> > > Thank you for your thoughtful response and the score adjustment! We appreciate your recognition of our efforts in clarifying the contributions and conducting the depth impact analysis. We will work on expanding the discussion of the related work and adding a detailed explanation of the experimental results in the revised version. Your constructive comments have been invaluable in improving our paper.

---

### Official Review · Reviewer_YRJg · 2024-11-04

**Soundness:** 2
**Presentation:** 3
**Contribution:** 2
**Rating:** 6
**Confidence:** 4

**Summary:**

This work proposes Wold-Model-Augmented (WMA) web agents. The authors present detailed analysis on the inability of current LLMs in estimating action consequences on the web and propose to abstract webpage state presentation as state transition description in natural language accordingly. Based on the abstraction, a world model for web navigation is trained and used for enhancing web agents. On two popular benchmarks, WebArena and Mind2Web, the proposed WMA web agent shows sizable performance gain over the baseline LLM and demands lower cost than tree-search methods.

**Strengths:**

1. To the best of my knowledge, this work is among the first to employ the world model in web agents. Its technical contribution includes a novel state representation, world model training method, and world model augmented agent design.
2. The resulting method shows improved performance and higher efficiency over baselines.
3. The experiment setup is overall reasonable. The authors provide sufficient ablation study and detailed analysis on performance, efficiency, and error distribution, contributing valuable insights to the community.
4. The writing and presentation of this paper is clear and easy to follow.

**Weaknesses:**

1. The experiments only cover end-to-end evaluation of agent performance. A directly evaluation on world modeling performance would be helpful for understanding the quality of the world model itself. For example, this evaluation can follow the same setting as the analysis of section 3.
2. The performance of Vanilla CoT and WMA web agent w/o policy optimization differs (13.1% vs 12.8% in Table 1). What are the differences between them? In addition, WMA web agent underperforms the tree search agent. Although tree search leverages additional signals and more compute, I think it is important to discuss if WMA web agent could scale to higher compute and deliver competitive performance under similar budget.
3. The proposed method and evaluation is limited to the text modality. Recent studies have suggested that visual features are critical for good web navigation performance. It remains unclear if WMA could adapt to multimodal settings and advance the state-of-the-art of web navigation.

**Questions:**

1. In section 4.1.1, training data for the world model is sampled from an LLM as web agent. Does this strategy create a dependency between the world model and the underlying agent, which could potentially hinder generalization?
2. The proposed method, particularly the state abstraction method, is limited to the text modality (HTML and accessibility tree). Can it be easily extended into a multimodal setup?
3. As stated in weaknesses, clarification of Vanilla CoT and WMA web agent /o policy optimization is needed.

---

> ### Author Response · Authors · 2024-11-28
>
> We appreciate your insightful review and acknowledgment of our contribution.
> ### **W1. In-depth analysis on the world model**
>
>
> We conducted a direct evaluation to evaluate how well our world model predicts the next observation, focusing primarily on coverage. Coverage measures how much information from the ground-truth observation is successfully captured in the model's prediction.
> We define coverage as the ratio of the information of the ground-truth observation covered by the predicted next observation. Specifically, the coverage score is calculated as:
>
> - (\# sentences in ground-truth observation covered by the predicted observation) / (\# total sentences in ground-truth observation).
>
>
>
>
> For evaluation, we employ an LLM-as-judge approach and we use GPT-4o as the judge LLM. We begin by separating both predicted and ground-truth observations into sentences. Then, we use an LLM to determine whether information from the target sentence can be found in the source text, which consists of a list of sentences.  We run the evaluation on 100 samples used in our preliminary analysis and compare our world model with a few different LLMs, including GPT-4o-mini and GPT-4o.
>
> |  | Coverage (%) |
> | :---- | :---- |
> | GPT-4o-mini | 33.50 |
> | GPT-4o | 33.85 |
> | Ours | 42.99 |
>
> We find that 42.99% of the information in the ground-truth observation is covered by our world model, and there is a significant gap between general LLMs that did not learn environment dynamics.
>
> ### **W3 (Q2). Extending the world model to take multimodal input**
> Thank you for your suggestion. We highly agree that the incorporation of both text and visual features is crucial for successful web navigation. Thus, we expanded our world model to take multimodal input (i.e., text-based observation and image-based observation) and we provide the results in the general response.
>
> ### **W2-1 (Q3). Clarification on the experimental setup**
> We assume the different performance of the w/o policy optimization setting and the reported Vanilla CoT stems from the number of max action limits, which was 5 for w/o policy optimization (SR 12.8%) and 30 for Vanilla CoT (SR 13.1%). We follow the setting of Tree Search Agent which also uses the max action limits as 5\. As mentioned in the Tree Search Agent paper, an action limit of 5 is a much more difficult setting for the web agents compared to a limit of 30\.
>
> ### **W2-2. Future potential of WMA web agent**
> Through the experiments in Figure 6 and in the general response, we find that scaling up the budget for the exploration and extending the world model to be capable of processing multimodal input can significantly improve the performance of our method. Also, we believe that scaling can close the performance gap between the tree search agent and our method and we are working on the experiment. We will share the results as soon as we get them.
>
>
> **Q1.** We agree with your concern. While we collect the exploration data with GPT-4o-mini, we find that the world model trained with the exploration data from GPT-4o-mini also works with GPT-4o agent as we show in Table 1 (a notable improvement from the Vanilla CoT method). Also, while the world model is trained with the Vanilla CoT agent’s trajectory, we show that our world model can be applied to other agent frameworks, such as SteP (in general response) and AWM.

---

> > ### Author Response · Authors · 2024-12-02
> >
> > Dear Reviewer YRJg,
> >
> > Thank you for your thoughtful review! Since we're in the last day of discussions, I was wondering if you could kindly take a look at our response? I'd love to know if it clarifies your questions.

---

> > > ### Comment · Reviewer_YRJg · 2024-12-02
> > >
> > > Thank you for your detailed responses and I'll maintain my positive score. Please include the additional content you presented during rebuttal in the revised version.

---

### Official Review · Reviewer_mAfj · 2024-11-04

**Soundness:** 3
**Presentation:** 2
**Contribution:** 2
**Rating:** 6
**Confidence:** 4

**Summary:**

This paper explores using explicit world models in LLM-based web agents. An (explicit) world model is a model able to predict the future state given an action on the current state. Their preliminary analysis shows that existing models cannot successfully predict the results of their actions, but that knowing that information would significantly help LLM-based agents.

Their contributions are
1) a training algorithm for web-agent world models that predicts an abstracted version of the transition to the observation after an action is applied.
2) an algorithm for using the world model to improve performance.

The training algorithm uses trajectory data to produce observation diffs from each action, then summarizes these changes with an LLM. The world model LLM is then trained to output the summaries (predict what will change in natural language)

The policy optimization algorithm uses a trained value function in addition to the world model. It works by sampling multiple potential actions, using the world model to predict changes, and then using the value function to predict the value. This can be thought of as a two-step Q-value function where the first step is the world model and the second step is evaluation. The value function is trained using reward data from Mind2Web.

They have main results on WebArena and Mind2Web. There is also further analysis by category, and other experiments, including ablations over the reward estimation, training of world model, and use of abstracted observations.

**Strengths:**

Originality:
The paper provides an original algorithm for training web-agent world models and an original algorithm for using such models to improve LLM-based web agents.

Clarity:
The clarity of the problem domain is quite good. The introduction lays a good case for how world models could improve web agents. The approach is quite clear. Figure 3 is nice. There is some breakdown of the clarity for some of the results but overall it was easy to follow along and understand what was being done.

Significance:
The utility of world models for web-agents is quite clear but still under-explored. Both the training and inference time algorithm presented here could have significance down the line.

**Weaknesses:**

The primary weakness of this paper is that the main results are quite weak. On WebArena, this significantly underperforms the listed baselines, and other newer baselines also exist.

The second weakness is that it is not always clear what is being tested or shown. There are a number of questions (see below) that need to be clarified.

To me the paper should focus more on the web-agent *world model* than on the web agent itself. There is some analysis on failure modes but not enough analysis on the world model itself. This may stem from the fact that abstract transition representations may be hard to analyze. A full world model would be able to show things like per-element accuracy. Perhaps there are other creative ways to analyze how to assess the learned world model and how it functions.

The paper really dives into this one way of using the world model when it could take a broader view. For instance, why not use WMA/policy optimization

**Questions:**

Clarifying questions:

1) What is “policy optimization” as applied to Tree Search Agent (Kuh et al, 2024). I thought Figure 3 bottom shows “policy optimization” which seems like it would only apply to World Model Agent.

2) Correspondingly, what is "no policy optimization"? Is that just CoT? If so, why are the results different? Why is Tree Search Agent better?

3) Is value function trained with the world model data or is it a pure value function? If pure value function, how does that work with the abstracted observations?

4) Line 284: I don’t get this line and how the citation matches. “explore diverse next states st+1 ∈ S (Wang et al., 2022)”

5) Table 5: The value function is fine-tuned. Is the Q-value function being fine-tuned as well and in the same way? If not, does not seem like a fair ablation. Also, is the Q-value function given ability to reason about what the action will do before outputting the value?
Other questions:

6) Why is the user instruction included in the data? Shouldn't the world model be able to predict the changes based on the action without needing to know the intent?

7) Table 2: Why not include the baselines from table 1? At least the tree search agent which is repeatedly compared to.

8) Figure 1 shows that commercial LLMs cannot predict the next state well by looking at actual observations. Perhaps they would be able to pick the correct abstracted transition-focused observation? Running that same experiment with the transition focused observation version (ground truth, not predicted) could be interesting.

9) Which leads me to the following question, how do larger commercial LLMs fare at predicting the next abstract transition? It would be nice to see Table 5 row 2 with a few different base LLMs.

10) Why is HTML-T5 considered SOTA for Mind2Web? It seems like there are many much stronger algorithms presented in table 1. Mind2Web is also an offline evaluation dataset so does not account for different ways of completing a task or different action spaces.

Other notes:

11) Line 184: “These findings highlight the necessity of world models in LLM-based web agents”:
- Not sure I agree with this. I would say something more like, “This findings highlight the benefit world models could provide to LLM-based web agents”.
- The findings show that (my words) “LLMs do not have good world models for web activity” and “having a better world model would improve web agent performance”. This does not add up to “necessity”

12) “In Table 1 (middle), we first compare our WMA web agent (16.6%) with vanilla CoT (13.1%) and observe significant improvements over almost all domains in WebArena as detailed in Table 2.”
This is confusingly written. Table 1 is the main table and there is only half a sentence of analysis.

13) I do not like the y-axis bounds selection on many of the figures. It would be easier to read with 0 as lower bound (or at least a bit more space). E.g. Figure 6 is hard to tell what the k=1 line is at.

---

> ### Author Response · Authors · 2024-11-26
> **Response to Reviewer mAfj - 1**
>
> We sincerely appreciate your detailed feedback, which is invaluable in improving our work.
>
> ### **Weaknesses**
>
> **W1. Performance of WMA web agents on WebArena**
> We design the framework to be easily applied to various agent frameworks, and in the experiments on WebArena we applied our framework to the most basic web agent, Vanilla CoT to understand its fundamental effectiveness.
>
> We also show that our approach also holds effectiveness with stronger agent frameworks (e.g., AWM) than Vanilla CoT as we show in Table 3\. Thus, while we did not achieve state-of-the-art (SOTA) performance on WebArena, we think that we can expect some performance improvement when it is adapted to the SOTA method, such as AWM.
>
> **W2. Need clarification on the experimental setup**
> We provide the answers to your questions below. Thank you again for the detailed review and suggestions\!
>
> **W3. In-depth analysis on the world model.**
>
> We conduct in-depth analyses to evaluate how well our world model predicts the next observation, focusing primarily on coverage. Coverage measures how much information from the ground-truth observation is successfully captured in the model's prediction.
> We define coverage as the ratio of the information of the ground-truth observation covered by the predicted next observation. Specifically, the coverage score is calculated as:
>
> - (\# sentences in ground-truth observation covered by the predicted observation) / (\# total sentences in ground-truth observation).
>
>
>
> For evaluation, we employ an LLM-as-judge approach using GPT-4o as the judge LLM. We begin by separating both predicted and ground-truth observations into sentences. Then, we use an LLM to determine whether information from the target sentence can be found in the source text, which consists of a list of sentences.  We run the evaluation on 100 samples used in our preliminary analysis and compare our world model with a few different LLMs, including GPT-4o-mini and GPT-4o.
>
> |  | Coverage (%) |
> | :---- | :---- |
> | GPT-4o-mini | 33.50 |
> | GPT-4o | 33.85 |
> | Ours | 42.99 |
>
> We find that 42.99% of the information in the ground-truth observation is covered by our world model, and there is a significant gap between general LLMs that did not learn environment dynamics.
>
> **W4. Other ways of using world models in web navigation.**
> We thought that there are two major ways to improve agents' policies during inference time. The first way is to sample multiple action candidates and select the best action (it can be regarded as BoN sampling), which is used as our method, and the second way is to refine the predicted action using the feedback.
>
> We have also explored the second way in Section 6.1. We run experiments on the Map domain from WebArena, and we find that the improvement is not that significant compared to the first way (WMA web agent: 22.3% and Self-refine w/ simulated feedback: 13.4%).

---

> ### Author Response · Authors · 2024-11-26
> **Response to Reviewer mAfj - 2**
>
> ### **Clarifying questions**
>
> > What is “policy optimization” as applied to Tree Search Agent (Koh et al, 2024). I thought Figure 3 bottom shows “policy optimization” which seems like it would only apply to World Model Agent.
>
> **Q1.** We use “policy optimization” to incur a method that selects the best action among the sampled actions according to the value score on the future states visited during the exploration. We have changed the wording to “action selection” for better clarity.
>
> ---
>
>
> > Correspondingly, what is "no policy optimization"? Is that just CoT? If so, why are the results different? Why is Tree Search Agent better?
>
> **Q2.** Yes, it is a setting that just uses CoT. We think that the results of CoT baseline with max action step 5 (average Success Rate 15.0%) from Tree Search Agent paper are too high, considering that CoT result with max action 30 is reported as 13.1% in the original WebArena paper, which is an easier setting as it allows more attempts.
>
> ---
>
>
> > Is value function trained with the world model data or is it a pure value function? If pure value function, how does that work with the abstracted observations?
>
> **Q3**. To finetune the value function, we transformed the original full observation into an abstract observation to apply to our world model. We will prepare a separate section in the Appendix to elaborate on the training details of the value function.
>
> ---
>
>
> > Line 284: I don’t get this line and how the citation matches. “explore diverse next states st+1 ∈ S (Wang et al., 2022)”
>
> **Q4.** We agree that this sentence can mislead the readers. We intended that a diverse exploration of the next states might help the agent to find the best path compared to sticking to a single path. We thought it was similar to generating diverse reasoning paths in Wang et al., 2022. We will revise this sentence to improve clarity. A revised sentence might be: "*We begin by sampling $k$ action candidates $\{a_t^1, a_t^2, ..., a_t^k\}$ from $\theta$ via top-$p$ decoding, to conduct diverse exploration on future observations $\{o_{t+1}^1, o_{t+1}^2, ..., o_{t+1}^k\}$ similar to Koh et al. (2024).*"
>
> ---
>
>
> > Table 5: The value function is fine-tuned. Is the Q-value function being fine-tuned as well and in the same way? If not, does not seem like a fair ablation. Also, is the Q-value function given ability to reason about what the action will do before outputting the value?
>
> **Q5.** Yes, it is fine-tuned as well in the same way. Also, we trained a Q-value function to explicitly generate rationales toward the values of which labels are annotated in the same way as our value function.

---

> > ### Comment · Reviewer_mAfj · 2024-12-02
> >
> > I am not seeing the new section on training the value function. I am confused because the value function has a prompt that encourages outputting text (thoughts) and a score. Can you elaborate a little further on the value functioning training objective here?

---

> ### Author Response · Authors · 2024-11-26
> **Response to Reviewer mAfj - 3**
>
> ### **Other Questions**
> > Why is the user instruction included in the data? Shouldn't the world model be able to predict the changes based on the action without needing to know the intent?
>
> **Q6.** Yes, by the definition of the world model, the user intent is not a necessary component of the input. However, in our framework, the goal of the world model is to provide the predicted future observation to the value function, so we thought that the description focused on the user intent might help the value function better understand the observation.
>
> ---
>
>
> > Table 2: Why not include the baselines from table 1? At least the tree search agent which is repeatedly compared to.
>
> **Q7.** It was hard to reproduce Tree Search Agent, as it requires around $1600 to run 812 instances in WebArena even if we use the cheapest commercial VLM, GPT-4o-mini.
>
> ---
>
>
> > Figure 1 shows that commercial LLMs cannot predict the next state well by looking at actual observations. Perhaps they would be able to pick the correct abstracted transition-focused observation? Running that same experiment with the transition-focused observation version (ground truth, not predicted) could be interesting.
>
> **Q8.** We additionally run preliminary analysis 1 (Figure 1\) with the transition-focused observation as you suggested. We find that the overall performances have increased, but still there is a gap between human performance, 0.83 .
>
> |  | GPT-4o-mini | GPT-4o | GPT-4-Turbo | Claude-3.5-Sonnet |
> | :---- | :---- | :---- | :---- | :---- |
> | **Full Observation** | **0.54** | **0.55** | **0.58** | **0.52** |
> | **Transition-focused Observation** | **0.62** | **0.66** | **0.70** | **0.61** |
>
> ---
>
>
> > Which leads me to the following question, how do larger commercial LLMs fare at predicting the next abstract transition? It would be nice to see Table 5 row 2 with a few different base LLMs.
>
> **Q9.** We run the experiment with a few different base LLMs and the results are in the table below. We find that the performance increases by using a stronger base LLM as the world model, but the ability to predict the next observation is still limited compared to our world model.
>
> |  | Llama-3.1-8B  | GPT-3.5-Turbo | GPT-4o-mini | WMA |
> | :---- | :---- | :---- | :---- | :---- |
> | **Shopping** | **14.0** | **8.0** | **30.0** | **32.0** |
> | **Gitlab** | **6.0** | **8.0** | **10.0** | **14.0** |
> | **Map** | **14.0** | **13.0** | **15.0** | **21.0** |
> | **Overall** | **12.0** | **10.5** | **17.5** | **22.0** |
>
> ---
>
>
> > Why is HTML-T5 considered SOTA for Mind2Web? It seems like there are many much stronger algorithms presented in table 1\. Mind2Web is also an offline evaluation dataset so does not account for different ways of completing a task or different action spaces.
>
> **Q10.** While we acknowledge that T5 itself is a relatively outdated model, HTML-T5 has a distinct advantage: it was specifically pretrained to process HTML format as input, enhancing web agents' environmental perception. This is similar to the recently released UGround, which uses massive visual input training to improve perception.
>
>
> The choice of baselines in Mind2Web is constrained by its offline nature. Many advanced baselines (such as AutoEval and Tree Search Agent) rely on verification/evaluation of environmental interaction outcomes, which isn't possible in Mind2Web's offline setting. This limitation led us to include AWM as an alternative baseline.
>
> ---
>
>
> **Q11,12,13 (other notes).** Thank you for the suggestions. We will update our draft accordingly.

---

> ### Comment · Area_Chair_PycE · 2024-11-27
>
> Dear reviewer mAfj,
>
> Thank you for your efforts reviewing this paper. Can you please check the authors' responses and see if your concerns have been addressed? Please acknowledge you have read their responses. Thank you!

---

> > ### Author Response · Authors · 2024-12-02
> >
> > Dear Reviewer mAfj,
> >
> > Thank you again for your time and effort in writing the review. As this is the final day of the discussion period, I would greatly appreciate if you could read our response to check it addresses your questions.

---

> > > ### Comment · Reviewer_mAfj · 2024-12-02
> > >
> > > I am so sorry for the delay! Thank you for your responses.  I am mostly satisfied with the responses provided the authors clarify them in the paper itself and include the presented data. There is still some question on the value function which hopefully the authors will be able to answer short notice.
> > >
> > > The main weakness is still the utility. That the model works with AWM is some evidence that it could help with state of the art models. The additional data presented here (and to other reviewers does improve the quality of the paper). I will adjust my score to weak accept but (like other reviewers), hope to see these requested and proposed changes/data in the camera ready.
> > >
> > > The draft still has some issues but given the short remaining discussion, I will give benefit of the doubt for camera ready. Please make sure to fix the y axes on figures so that they are not just the min and max of the data range. e.g. 0 as min. I have updated the presentation score to fair.

---

> > > > ### Author Response · Authors · 2024-12-03
> > > >
> > > > Thank you for the reply and the score adjustment. We will **make sure** to include the following in the camera-ready version:
> > > > - the presented experimental results in our response
> > > > - adjustments of the y-axis of the figures
> > > > - clarifications on the experimental settings (including the value function)
> > > >
> > > > ### Explanations of the Value Function
> > > > We explored two settings, prompting and fine-tuning, to implement the value function.
> > > > For fine-tuning the setting, we use the next-token prediction objective for the training. While the general way to implement reward models is to use an auxiliary regression head, instead of the LM head, the generative manner allows LLMs to output their thoughts on the evaluation. Recently, there has been a line of research [1,2,3] that has approached reward modeling in a generative manner, to fully leverage LLMs' reasoning abilities by explicitly outputting the thoughts on the evaluation. Following, this line of research, we train the value function in a generative manner so that it can explicitly generate the implicit rationales before the scores.
> > > >
> > > > For the prompted value function, we use the same prompt used for the fine-tuned value function to encourage them to output both the thoughts and the scores.
> > > >
> > > > [1] Generative Verifiers: Reward Modeling as Next-Token Prediction
> > > >
> > > > https://arxiv.org/pdf/2408.15240
> > > >
> > > > [2] Generative Reward Models
> > > >
> > > > https://arxiv.org/pdf/2410.12832
> > > >
> > > > [3] Foundational Autoraters: Taming Large Language Models for Better Automatic Evaluation
> > > >
> > > > https://arxiv.org/pdf/2407.10817

---

### Official Review · Reviewer_N8Nf · 2024-11-08

**Soundness:** 2
**Presentation:** 3
**Contribution:** 3
**Rating:** 6
**Confidence:** 4

**Summary:**

The paper proposes an approach for enhancing the performance of LLM-based web agents in long-horizon tasks. The authors introduce a World-Model-Augmented (WMA) web agent that simulates the outcomes of its actions through a "world model." This design enables the agent to anticipate the effects of actions, thus reducing errors and improving decision-making in dynamic web environments. By employing a transition-focused observation abstraction, the model processes large state transitions without redundant data. Experiments conducted on WebArena and Mind2Web indicate that the WMA agent outperforms other agents in various metrics.

**Strengths:**

1. The paper considers the application of world models within LLM-based web agents, creating an interesting direction for improving policy selection in web-based navigation tasks.

2. The proposed transition-focused observation abstraction helps to reduce redundant data, lowering computational costs and allowing the model to focus on critical changes.

3. The authors show that the WMA model can be easily integrated into existing web agents, making it a versatile addition to LLM-based navigation models.

**Weaknesses:**

1. While the paper demonstrates improved single-step action selection, it does not address multi-step planning comprehensively, which is crucial for certain long-horizon tasks.

2. The approach is limited to text-based web interactions, and although HTML trees are effective, omitting visual elements could restrict the model's applicability in web navigation tasks where visual cues are essential.

3. While the transition-focused abstraction reduces redundancy, it may oversimplify complex web elements, potentially leading to reduced accuracy in actions involving nuanced page layouts.

4. The paper does not discuss how the proposed world model scales when dealing with significantly larger action spaces, which is critical for broader applicability.

**Questions:**

1. Can the model extend to handle visual inputs alongside text-based observations? This would make it more versatile for environments where visual information is vital.

2. How does the model perform in extended multi-step planning scenarios? Would recursive application of the world model effectively address multi-step planning challenges?

3. How does the observation abstraction perform in very dynamic web environments? In fast-changing states, does it retain relevant information, or does it risk omitting crucial details?

4. What is the impact of scaling the action space on the computational efficiency of the WMA model? Would the model maintain its efficiency with a larger action candidate set?

---

> ### Author Response · Authors · 2024-11-27
> **Response to Reviewer N8Nf**
>
> Thank you for the positive review and valuable comments.
>
> ### **W1(Q2).** **Application of world model to multi-step planning**
>
> We agree that extending our framework to a multi-step setting is crucial for web navigation tasks since web navigation generally involves long-horizon planning. To explore this direction, we conduct an experiment and provide the results in the general response.
>
> ### **W2 (Q1). Application of world model to multimodal input**
> We have extended our world model to take not only text observation but also multimodal observation. These results are also added in the general comment.
>
> ### **W3 (Q3). Human evaluation on the quality of transition-focused observation abstraction**
>
> To assess the quality of transition-focused abstraction, we conducted a human evaluation using Amazon Mechanical Turk raters. We first ranked instances in the training set based on the magnitude of changes between consecutive observations ($\\Delta(o\_t, o\_{t+1})$) and divided them into four quantiles:
>
> * (25%, 0%\]: Largest changes between observations
> * (50%, 25%\]
> * (75%, 50%\]
> * \[100%, 75%\]: Smallest changes between observations
>
> We randomly sampled 50 instances from each group and asked 3 different raters per each instance to evaluate how well the abstraction preserved important information using a 5-point Likert scale, where:
>
> * Score 5: Fully covers all important information.
> * Score 4: Covers most important information, with minor omissions.
> * Score 3: Covers some important information, but leaves out some detail.
> * Score 2: Covers very little information, with major omissions.
> * Score 1: Lacks most of the important information.
>
> | \# lines of $\Delta(o_t, o_{t+1})$ | Top \[100%,75%\] | Top (75%, 50%\] | Top (50%,25%\] | Top (25%,0%\] |
> | :---- | :---- | :---- | :---- | :---- |
> | Score | 3.91 | 3.85 | 3.84 | 4.06 |
>
> The human evaluation results demonstrate that our transition-focused abstraction strategy effectively preserves important information, even in highly dynamic situations. While we observe a slight decrease in scores as transitions become more dynamic in Top (75%, 50%\] and Top (50%, 25%\], the group with the largest changes (Top (25%, 0%\]) still maintains a strong score close to 4, which means most of the important information is preserved. These results validate that our approach successfully captures essential information despite significant changes between observations.
>
> ### **W4 (Q4).** **How WMA web agents scale with larger action candidates**
> We conducted an additional analysis to examine how increasing the number of action candidates ($k$) affects performance and computational efficiency. Since we adopted the Tree Search Agent codebase for implementing action sampling logic, action candidates are sampled in the following manner:
>
> 1. Generate $max(20,  k*2)$ output sequences.
> 2. Build a counter dictionary consisting of (action candidate, frequency) to remove identical actions and rank actions with their probability.
> 3. Select top-$k$ actions from the dictionary.
>
> Thus, the API cost for sampling action remains unchanged until $k\<=10$.
>
> The table below shows how the number of action candidates ($k$) affects both the performance and computational cost of the WMA web agent on the Map domain from WebArena. As mentioned above, API costs remain similar for $k≤10$ but increase when $k\>10$. Meanwhile, the inference time, which is primarily bounded by environment interaction time, shows only a gradual increase as $k$ increases.
>
> |  | k=1 | k=3 | k=5 | k=7 | k=9 | k=20 |
> | :---- | ----- | ----- | ----- | ----- | ----- | ----- |
> | Success Rate | 7.1 | 17.0  | 16.1 | 21.4 | 20.5 | 22.0 |
> | API Cost | 0.11 | 0.12 | 0.12 | 0.12 | 0.12 | 0.36 |
> | Time | 126.7 | 123.9 | 126.2 | 128.3 | 134.4 | 138.4 |

---

> > ### Author Response · Authors · 2024-12-02
> >
> > Dear Reviewer N8Nf,
> >
> > Thank you for your thorough review of our manuscript. Your feedback has been invaluable in improving our work. As we have reached the final day of the discussion period, we would be grateful if you could review our response to check if it adequately addresses your questions.

---

### Author Response · Authors · 2024-11-26
**General Response**

First of all, thank you for your patience. We have dedicated the entire discussion period to conducting experiments in response to the reviewers' insightful and interesting suggestions, as we wanted to thoroughly address their requests.
We provide the experiments that the reviewers commonly mention in this general response.

### **Extending our world model to take multimodal input**

Specifically, we enhanced the model to process both text and image observations. We extended our world model to a multimodal setting, inspired by the recent success of multimodal web agents. For our experiments, we used the Mind2Web (cross-task) dataset with Qwen-2-VL-2B as the backbone Vision Language Model.

Despite using a smaller parameter size compared to Llama-3.1-8B, the multimodal input led to notable improvements across all metrics. These results demonstrate two key findings: (1) our framework can readily adapt to multimodal settings, and (2) the addition of visual modality provides clear benefits. Given that we used a naive approach to image input, we expect further improvement when incorporating more sophisticated image prompting techniques, such as SeeAct or Set-of-Marks, could further enhance performance.


|  | Modality of World Model | Elem Acc | OF1 | Step SR | SR |
| :---- | ----- | ----- | ----- | ----- | ----- |
| MindAct | \- | \- |\-  | 17.4 | 0.8 |
| AWM | \- | 78.3 | 74.1 | 62.8 | 15.3 |
| AWM+WMA (Llama-3.1-8B-textonly) | Text | 79.9 | 75.8 | 67.0 | 25.4 |
| AWM+WMA (Qwen-VL-2B-textonly) | Text | 79.2 | 75.1 | 65 | 23.7 |
| AWM+WMA (Qwen-VL-2B-multimodal) | Text+Image | **83.0** | **78.9** | **72.8** | **36.7** |

### **Extending WMA web agent to multi-step exploration settings (depth + width)**
Beyond searching for the best action in a single step, we also explored finding optimal paths through multi-step search by iteratively using the world model. However, prediction errors accumulate as search depth increases when using only world model simulations, limiting real-world applicability. To address this, we adopt a hybrid approach combining simulated interactions for width exploration and actual environmental interactions for depth exploration.


We used A*-like search algorithm following Tree Search Agent and conducted experiments using the same settings as shown in Figure 6 (Ablation on the number of sampled actions), with results presented in the table below. We found that increasing the depth from (w=3, d=1) to (w=3, d=3) improves performance (17.1 -> 19.6). However, when comparing settings with the same exploration budget - (w=9, d=1) vs. (w=3, d=3), we find that allocating more budget to width shows slightly better performance.

A specific challenge explains this issue: Many errors occur during the execution of optimal action sequences due to mismatches between search time and execution time. Even when a browser loads the same content from the same URL, element IDs calculated by the backend can change upon page reload. This means that the same element might have different IDs between the search and execution phases.


| width (w) /depth (d) | d=1 | d=2 | d=3 |
| :---- | :---- | :---- | :---- |
| **w=1** | **7.1** | **\-** | **10.7** |
| **w=2** | **10.1** | **12.6** | **\-** |
| **w=3** | **17.1** | **\-** | **19.6** |
| **w=9** | **20.5** | **\-** | **\-** |

---

### Meta-Review · Area_Chair_PycE · 2024-12-24

**Metareview:**

Summary:

This paper presents a World-model-augmented (WMA) web agent, which simulates the outcomes of the agent’s actions for better decision-making. To train a world model for web agents, the paper proposes a transition-focused observation abstraction, where the prediction objectives are free-form natural language descriptions exclusively highlighting important state differences between time steps.  This helps overcome the challenges of training LLMs as world models predicting next observations, such as repeated elements across observations and long HTML inputs. Experiments on WebArena and Mind2Web show that the trained world model can improve agents' policy selection. Compared with tree search agents, the WMA agent demonstrates cost- and time-efficiency.

Strengths:

1. The study of world models within LLM-based web agents is an interesting direction. The paper provides a new method for training world models for web agents and a new method for using such models to improve LLM-based web agents. (Reviewer N8Nf, Reviewer mAfj) The paper’s technical contribution includes a novel state representation, world model training method, and world model augmented agent design. (Reviewer YRJg)

2. The utility of world models for web-agents is still under-explored. Both the training and inference time algorithm presented in this paper could have significance down the line. (Reviewer mAfj)).

3. The WMA agent shows improved efficiency over the tree search agent baseline (Reviewer YRJg).

4. In general, the writing of the paper is easy to follow. (Reviewer YRJg, tKJR)

Weaknesses:

1. Lack of experiments to show the effectiveness of the world model under a multimodal setup (especially how it would improve a multimodal agent). The authors seem to have only considered taking multimodal information as input to the world model itself during the rebuttal period, but did not experiment with incorporating the world models into more recent multimodal agents.

2. The proposed WMA agent does not consistently outperform baselines in terms of SR (e.g., WMA web agent vs. tree search agent in Table 1). Its effectiveness is not entirely convincing.

**Additional Comments On Reviewer Discussion:**

Many issues have been addressed during the discussion period, including:

1. Differences from previous work, such as “Agent Planning with World Knowledge Model”, have been clarified.

2. Concerns regarding lack of experiments on the impact of the width and depth have been addressed.

---

### Decision · Program_Chairs · 2025-01-22

Accept (Poster)